# Bandit Learning with Positive Externalities

**Virag Shah**
Management Science and Engineering
Stanford University
California, USA 94305
virag@stanford.edu

**Jose Blanchet**
Management Science and Engineering
Stanford University
California, USA 94305
jblanche@stanford.edu

**Ramesh Johari**
Management Science and Engineering
Stanford University
California, USA 94305
rjohari@stanford.edu

## Abstract

In many platforms, user arrivals exhibit a self-reinforcing behavior: future user arrivals are likely to have preferences similar to users who were satisfied in the past. In other words, arrivals exhibit *positive externalities*. We study multiarmed bandit (MAB) problems with positive externalities. We show that the self-reinforcing preferences may lead standard benchmark algorithms such as UCB to exhibit linear regret. We develop a new algorithm, Balanced Exploration (BE), which explores arms carefully to avoid suboptimal convergence of arrivals before sufficient evidence is gathered. We also introduce an adaptive variant of BE which successively eliminates suboptimal arms. We analyze their asymptotic regret, and establish optimality by showing that no algorithm can perform better.

## 1 Introduction

A number of different platforms use multiarmed bandit (MAB) algorithms today to optimize their service: e.g., search engines and information retrieval platforms; e-commerce platforms; and news sites. Many such platforms exhibit a natural self-reinforcement in the arrival process of users: future arrivals may be biased towards users who expect to have positive experiences based on the past outcomes of the platform. For example, if a news site generates articles that are liberal (resp., conservative), then it is most likely to attract additional users who are liberal (resp., conservative) [2]. In this paper, we study the optimal design of MAB algorithms when user arrivals exhibit such positive self-reinforcement.

We consider a setting in which a platform faces many types of users that can arrive. Each user type is distinguished by preferring a subset of the item types above all others. The platform is not aware of either the type of the user, or the item-user payoffs. Following the discussion above, arrivals exhibit *positive externalities* (also called positive network effects) among the users [13]: in particular, if one type of item generates positive rewards, users who prefer that type of item become more likely to arrive in the future.

Our paper quantifies the consequences of positive externalities for bandit learning in a benchmark model where the platform is unable to observe the user's type on arrival. In the model we consider, introduced in Section 3, there is a set of $m$ arms. A given arriving user prefers a subset of these arms over the others; in particular, all arms other than the preferred arms generate zero reward. A preferred arm $a$ generates a Bernoulli reward with mean $\mu_a$. To capture positive externalities, the probability

| | $\alpha = 0$ | $0 < \alpha < 1$ | $\alpha = 1$ | $\alpha > 1$ |
|---|---|---|---|---|
| Lower Bound | $\Omega(\ln T)$ | $\Omega(T^{1-\alpha} \ln^\alpha T)$ | $\Omega(\ln^2 T)$ | $\Omega(\ln^\alpha T)$ |
| UCB | $O(\ln T)$ | $\Omega(T)$ | $\Omega(T)$ | $\Omega(T)$ |
| Random-explore-then-commit | $O(\ln T)$ | $\Omega\left(T^{1-\alpha} \ln^{\frac{\alpha}{1-\alpha}} T\right)$ | $\Omega\left(T^{\frac{\mu_b}{\mu_b + \theta_{a*} \mu_{a*}}}\right)$ | $\Omega(T)$ |
| Balanced Exploration (BE) | $\tilde{O}(\ln T)$ | $\tilde{O}(T^{1-\alpha} \ln^\alpha T)$ | $\tilde{O}(\ln^2 T)$ | $\tilde{O}(\ln^\alpha T)$ |
| BE with Arm Elimination (BE-AE) | $O(\ln T)$ | $O(T^{1-\alpha} \ln^\alpha T)$ | $O(\ln^2 T)$ | $O(\ln^\alpha T)$ |

Table 1: Total regret under different settings. Here $a^* = \arg\max \mu_a$, and $b = \arg\max_{a \neq a^*} \mu_a$. For Random-explore-then-commit algorithm, we assume that the initial bias $\theta_a$ for each arm $a$ is a positive integer (cf. Section 3). The notation $f(T) = \tilde{O}(g(T))$ implies there exists $k > 0$ such that $f(T) = O(g(T) \ln^k g(T))$.

that a user preferring arm $a$ arrives at time $t$ is proportional to $(S_a(t-1) + \theta_a)^\alpha$, where $S_a(t-1)$ is the total reward observed from arm $a$ in the past and $\theta_a$ captures the initial conditions. The positive constant $\alpha$ captures the strength of the externality: when $\alpha$ is large the positive externality is strong.

The platform aims to maximize cumulative reward up to time horizon $T$. We evaluate our performance by measuring regret against an "offline" oracle that always chooses the arm $a^* = \arg\max_a \mu_a$. Because of the positive externality, this choice causes the user population to shift entirely to users preferring arm $a^*$ over time; in particular, the oracle achieves asymptotically optimal performance to leading order in $T$. We study the asymptotic scaling of cumulative regret against the oracle at $T$ as $T \to \infty$.

At the heart of this learning problem is a central tradeoff. On one hand, because of the positive externality, the platform operator is able to move the user population towards the profit maximizing population. On the other hand, due to self-reinforcing preferences the impact of *mistakes* is amplified: if rewards are generated on suboptimal arms, the positive externality causes more users that prefer those arms to arrive in the future. We are able to explicitly quantify the impact of this tradeoff in our model.

Our main results are as follows.

**Lower bound**. In Section 4, we provide an explicit lower bound on the best achievable regret for each $\alpha$. Strikingly, the optimal regret is structurally quite different than classical lower bounds for MAB problems; see Table 1. Its development sheds light into the key differences between MABs with positive externalities and those without.

**Suboptimality of classical approaches**. In Section 5, we show that the UCB algorithm is not only suboptimal, but in fact has positive probability of *never* obtaining a reward on the best arm $a^*$—and thus obtains linear regret. This is because UCB does not explore sufficiently to find the best arm. However, we show that just exploring more aggressively is also insufficient; a random-explore-then-commit policy which explores in an unstructured fashion remains suboptimal. This demonstrates the need of developing a new approach to exploration.

**Optimal algorithm**. In Section 6, we develop a new algorithmic approach towards optimizing the exploration-exploitation tradeoff. Interestingly, this algorithm is cautious in the face of uncertainty to avoid making long-lasting mistakes. Our algorithm, Balanced Exploration (BE), keeps the user population "balanced" during the exploration phase; by doing so, it exploits an arm only when there is sufficient certainty regarding its optimality. Its adaptive variant, Balanced Exploration with Arm Elimination (BE-AE), intelligently eliminates suboptimal arms while balancing exploration among the remainder. BE has the benefit of not depending on system parameters, while BE-AE uses such information (e.g., $\alpha$). We establish their optimality by developing an upper-bound on their regret for each $\alpha$; this nearly matches the lower bound (for BE), and exactly matches the lower bound (for BE-AE).

Further, in Section 7 we provide simulation results to obtain quantitative insights into the relative performance of different algorithms. We conclude the paper by summarizing the main qualitative insights obtained from our work.

## 2  Related work

As noted above, our work incorporates positive externalities in user arrivals. Positive externalities are also referred to as positive network effects or positive network externalities. (Note that the phrase "network" is often used here, even when the effects do not involve explicit network connections between the users.) See [13], as well as [21, 20] for background. Positive externalities are extensively discussed in most standard textbooks on microeconomic theory; see, e.g., Chapter 11 of [17].

It is well accepted that online search and recommendation engines produce feedback loops that can lead to self-reinforcement of popular items [3, 6, 19, 9]. Our model captures this phenomenon by employing a self-reinforcing arrival process, inspired by classical urn processes [4, 12].

We note that the kind of self-reinforcing behavior observed in our model may be reminiscent of "herding" behavior in Bayesian social learning [7, 23, 1]. In these models, arriving Bayesian rational users take actions based on their own private information, and the outcomes experienced by past users. The central question in that literature is the following: do individuals base their actions on their own private information, or do they follow the crowd? By contrast, in our model it is the platform which takes actions, without directly observing preferences of the users.

If the user preferences are known then a platform might choose to personalize its services to satisfy each user individually. This is the theme of much recent work on *contextual bandits*; see, e.g., [16, 22, 18] and [8] for a survey of early work. In such a model, it is important that either (1) enough observable covariates are available to group different users together as decisions are made; or (2) users are long-lived so that the platform has time to learn about them.

In contrast to contextual bandits, in our model the users' types are not known, and they are short-lived (one interaction per user). Of course, the reality of many platforms is somewhere in between: some user information may be available, though imperfect. We view our setting as a natural benchmark model for analysis of the impact of self-reinforcing arrivals. Through this lens, our work suggests that there are significant consequences to learning when the user population itself can change over time, an insight that we expect to be robust across a wide range of settings.

## 3  Preliminaries

In this section we describe the key features of the model we study. We first describe the model, including a precise description of the arrival process that captures positive externalities. Next, we describe our objective: minimization of regret relative to the expected reward of a natural oracle policy.

### 3.1  Model

**Arms and rewards.** Let $A = \{1, ..., m\}$ be the set of available arms. During each time $t \in \{1, 2, ...\}$ a new user arrives and an arm is "pulled" by the platform; we denote the arm pulled at time $t$ by $I_t$. We view pulling an arm as presenting the corresponding option to the newly arrived user. Each arriving user prefers a subset of the arms, denoted by $J_t$. We describe below how $J_t$ is determined.

If arm $a$ is pulled at time $t$ and if the user at time $t$ prefers arm $a \in A$ (i.e., $a \in J_t$) then the reward obtained at time $t$ is an independent Bernoulli random variable with mean $\mu_a$. We assume $\mu_a > 0$ for all arms. If the user at time $t$ does not prefer the arm pulled then the reward obtained at time $t$ is zero. We let $X_t$ denote the reward obtained at time $t$.

For $t \geq 1$, let $T_a(t)$ represent the number of times arm $a$ is pulled up to and including time $t$, and let $S_a(t)$ represent the total reward accrued by pulling arm $a$ up to and including time $t \geq 1$. Thus $T_a(t) = |\{1 \leq s \leq t : I_s = a\}|$, and $S_a(t) = |\{1 \leq s \leq t : I_s = a, X_s = 1\}|$. We define $T_a(0) = S_a(0) = 0$.

**Unique best arm**. We assume there exists a unique $a^* \in A$ such that:

$$a^* = \arg\max \mu_a.$$

This assumption is standard and made for technical convenience; all our results continue to hold without it.

**Arrivals with positive externalities.** We now define the arrival process $\{J_t\}_{t \geq 1}$ that determines users' preferences over arms; this arrival process is the novel feature of our model. We assume there are fixed constants $\theta_a > 0$ for $a \in A$ (independent of $T$), denoting the initial "popularity" of arm $a$.

For $t \geq 0$, define:

$$N_a(t) = S_a(t) + \theta_a, \quad a \in A.$$

Observe that by definition $N_a(0) = \theta_a$.

In our arrival process, arms with higher values of $N_a(t)$ are more likely to be preferred. Formally, we assume that the $t^{\text{th}}$ user prefers arm $a$ (i.e., $a \in J_t$) with probability $\lambda_a(t)$ independently of other arms, where:

$$\lambda_a(t) = \frac{f(N_a(t-1))}{\sum_{a'=1}^{m} f(N_{a'}(t-1))},$$

where $f(\cdot)$ is a positive, increasing function $f$. We refer to $f$ as the *externality function*. In our analysis we primarily focus on the parametric family $f(x) = x^\alpha$, where $\alpha \in (0, \infty)$.

Intuitively, the idea is that agents who prefer arm $a$ are more likely to arrive if arm $a$ has been successful in the past. This is a positive externality: users who prefer arm $a$ are more likely to generate rewards when arm $a$ is pulled, and this will in turn increase the likelihood an arrival preferring arm $a$ comes in the future. The parameter $\alpha$ controls the strength of this externality: the positive externality is stronger when $\alpha$ is larger.

If $f$ is linear ($\alpha = 1$), then we can interpret our model in terms of an urn process. In this view, $\theta_a$ resembles the initial number of balls of color $a$ in the urn at time $t = 1$ and $N_a(t)$ resembles the total number of balls of color $a$ added into the urn after $t$ draws. Thus, the probability the $t^{\text{th}}$ draw is of color $a$ is proportional to $N_a(t)$. In contrast to the standard urn model, in our model we have additional control: namely, we can pull an arm, and thus govern the probability with which a new ball of the same color is added into the urn.

### 3.2 The oracle and regret

**Maximizing expected reward.** Throughout our presentation, we use $T$ to denote the time horizon over which performance is being optimized. (The remainder of our paper characterizes upper and lower bounds on performance as the time horizon $T$ grows large.) We let $\Gamma_T$ denote the total reward accrued up to time $T$:

$$\Gamma_T = \sum_{t=1}^{T} X_t.$$

The goal of the platform is to choose a sequence $\{I_t\}$ to maximize $\mathbb{E}[\Gamma_T]$. As usual, $I_t$ must be a function only of the past history (i.e., prior to time $t$).

**The oracle policy.** As is usual in multiarmed bandit problems, we measure our performance against a benchmark policy that we refer to as the Oracle.

**Definition 1** (Oracle)**.** *The* Oracle *algorithm knows the optimal arm* $a^*$, *and pulls it at all times* $t = 1, 2, \ldots$.

Let $\Gamma_T^*$ denote the reward of the Oracle. Note that Oracle may not be optimal for finite fixed $T$; in particular, unlike in the standard stochastic MAB problem, the expected cumulative reward $\mathbb{E}[\Gamma_T^*]$ is not $\mu_{a^*}T$, as several arrivals may not prefer the optimal arm.

The next proposition provides tight bounds on $\mathbb{E}[\Gamma_T^*]$. For the proof, see the Appendix.

**Proposition 1.** *Suppose* $\alpha > 0$. *Let* $\theta^\alpha = \sum_{a \neq a^*} \theta_a^\alpha$. *The expected cumulative reward* $\mathbb{E}[\Gamma_T^*]$ *for the* Oracle *satisfies:*

*1.* $\mathbb{E}[\Gamma_T^*] \leq \mu_{a^*}T - \mu_{a^*}\theta^\alpha \sum_{k=1}^{T} \frac{1}{(k + \theta_{a^*} - 1)^\alpha + \theta^\alpha}.$

2. $\mathbb{E}[\Gamma_T^*] \geq \mu_{a^*} T - \theta^\alpha \sum_{k=1}^T \frac{1}{(k + \theta_{a^*})^\alpha} - 1.$

*In particular, we have:*

$$\mathbb{E}[\Gamma_T^*] = \begin{cases} \mu_{a^*} T - \Theta(T^{1-\alpha}), & 0 < \alpha < 1 \\ \mu_{a^*} T - \Theta(\ln T), & \alpha = 1 \\ \mu_{a^*} T - \Theta(1), & \alpha > 1 \end{cases}$$

The discontinuity at $\alpha = 1$ in the asymptotic bound above arrises since $\sum_{k=1}^T \frac{1}{k^\alpha}$ diverges for each $\alpha \leq 1$ but converges for $\alpha > 1$. Further, the divergence is logarithmic for $\alpha = 1$ but polynomial for each $\alpha < 1$.

Note that in all cases, the reward asymptotically is of order $\mu_{a^*} T$. This is the best achievable performance to leading order in $T$, showing that the oracle is asymptotically optimal.

**Our goal: Regret minimization.** Given any policy, define the *regret* against the Oracle as $R_T$:

$$R_T = \Gamma_T^* - \Gamma_T. \tag{1}$$

Our goal in the remainder of the paper is to minimize the expected regret $\mathbb{E}[R_T]$. In particular, we focus on characterizing regret performance asymptotically to leading order in $T$ (both lower bounds and achievable performance), for different values of the externality exponent $\alpha$.

## 4 Lower bounds

In this section, we develop lower bounds on the achievable regret of any feasible policy. As we will find, these lower bounds are quite distinct from the usual $O(\ln T)$ lower bound (see [15, 8]) on regret for the standard stochastic MAB problem. This fundamentally different structure arises because of the positive externalities in the arrival process.

To understand our construction of the lower bound, consider the case where the externality function is linear ($\alpha = 1$); the other cases follow similar logic. Our basic idea is that in order to determine the best arm, any optimal algorithm will need to explore all arms at least $\ln T$ times. However, this means that after $t' = \Theta(\ln T)$ time, the total reward on any suboptimal arms will be of order $\sum_{b \neq a^*} N_b(t') = \Theta(\ln T)$. Because of the effect of the positive externality, any algorithm will then need to "recover" from having accumulated rewards on these suboptimal arms. We show that even if the optimal arm $a^*$ is pulled from time $t'$ onwards, a regret $\Omega(\ln^2 T)$ is incurred simply because arrivals who do not prefer arm $a^*$ continue to arrive in sufficient numbers.

The next theorem provides regret lower bounds for all values of $\alpha$. The proof can be found in the Appendix.

**Theorem 1.**    1. *For $\alpha < 1$, there exists no policy with $\mathbb{E}[R_T] = o(T^{1-\alpha} \ln^\alpha T)$ on all sets of Bernoulli reward distributions.*

2. *For $\alpha = 1$, there exists no policy with $\mathbb{E}[R_T] = o(\ln^2 T)$ on all sets of Bernoulli reward distributions.*

3. *For $\alpha > 1$, there exists no policy with $\mathbb{E}[R_T] = o(\ln^\alpha T)$ on all sets of Bernoulli reward distributions.*

The remainder of the paper is devoted to studying regret performance of classic algorithms (such as UCB), and developing an algorithm that achieves the lower bounds above.

## 5 Suboptimality of classical approaches

We devote this section to developing structural insight into the model, by characterizing the performance of two classical approaches for the standard stochastic MAB problem: the UCB algorithm [5, 8] and a random-explore-then-commit algorithm.

## 5.1 UCB

We first show that the standard upper confidence bound (UCB) algorithm, which does not account for the positive externality, performs poorly. (Recall that in the standard MAB setting, UCB achieves the asymptotically optimal $O(\ln T)$ regret bound [15, 8].)

Formally, the UCB algorithm is defined as follows.

**Definition 2** (UCB($\gamma$))**.** *Fix $\gamma > 0$. For each $a \in A$, let $\hat{\mu}_a(0) = 0$ and for each $t > 0$ let $\hat{\mu}_a(t) := \frac{S_a(t-1)}{T_a(t-1)}$, under convention that $\hat{\mu}_a(t) = 0$ if $T_a(t-1) = 0$. For each $a \in A$ let $u_a(0) = 0$ and for each $t > 0$ let*

$$u_a(t) := \hat{\mu}_a(t) + \sqrt{\frac{\gamma \ln t}{T_a(t-1)}}.$$

*Choose:*

$$I_t \in \arg\max_{a \in A} u_a(t),$$

*with ties broken uniformly at random.*

Under our model, consider an event where $a^* \notin J_t$ but $I_t = a^*$: i.e., $a^*$ is pulled but the arriving user did not prefer arm $a^*$. Under UCB, such events are self-reinforcing, in that they not only lower the upper confidence bound for arm $a^*$, resulting in fewer future pulls of arm $a^*$, but they also reduce the preference of *future users* towards arm $a^*$.

It is perhaps not surprising, then, that UCB performs poorly. However, the impact of this self-reinforcement under UCB is so severe that we obtain a striking result: there is a strictly positive probability that the optimal arm $a^*$ will *never* see a positive reward, as shown by the following theorem. An immediate consequence of this result is that the regret of UCB is linear in the horizon length. The proof can be found in the Appendix.

**Theorem 2.** *Suppose $\gamma > 0$. Suppose that $f(x)$ is $\Omega\left(\ln^{1+\epsilon}(x)\right)$ for some $\epsilon > 0$. For UCB($\gamma$) algorithm, there exists an $\epsilon' > 0$ such that*

$$\mathbb{P}\left(\lim_{T\to\infty} S_{a^*}(T) = 0\right) \geq \epsilon'.$$

*In particular, the regret of UCB($\gamma$) is $O(T)$.*

## 5.2 Random-explore-then-commit

UCB fails because it does not explore sufficiently. In this section, we show that more aggressive unstructured exploration is not sufficient to achieve optimal regret. In particular, we consider a policy that chooses arms independently and uniformly at random for some period of time, and then commits to the empirical best arm for the rest of the time.

**Definition 3** (REC($\tau$))**.** *Fix $\tau \in \mathbb{Z}_+$. For each $1 \leq t \leq \tau$, choose $I_t$ uniformly at random from set $A$. Let $\hat{a}^* \in \arg\max_a S_a(\tau)$, with tie broken at random. For $\tau < t < T$, $I_t = a^*$.*

The following theorem provides performance bounds for the REC($\tau$) policy for our model. The proof of this result takes advantage of multitype continuous-time Markov branching processes [4, 12]; it is given in the Appendix.

**Proposition 2.** *Suppose that $\theta_a$ for each $a \in A$ is a positive integer. Let $b = \arg\max_{a \neq a^*} \mu_a$. The following statements hold for the REC($\tau$) policy for any $\tau$:*

*1. If $0 < \alpha < 1$ then we have $\mathbb{E}[R_T] = \Omega(T^{1-\alpha} \ln^{\frac{\alpha}{1-\alpha}} T)$.*

*2. If $\alpha = 1$ then we have $\mathbb{E}[R_T] = \Omega\left(T^{\frac{\mu_b}{\mu_b + \theta_{a^*} \mu_{a^*}}}\right).$*

*3. If $\alpha > 1$ then we have $\mathbb{E}[R_T] = \Omega(T)$.*

Thus, for $\alpha \leq 1$, the REC($\tau$) policy may improve on the performance of UCB by delivering sublinear regret. Nevertheless this regret scaling remains suboptimal for each $\alpha$. In the next section, we demonstrate that carefully structured exploration can deliver an optimal regret scaling (matching the lower bounds in Theorem 1).

# 6 Optimal algorithms

In this section, we present an algorithm that achieves the lower bounds presented in Theorem 1. The main idea of our algorithm is to structure exploration by *balancing* exploration across arms; this ensures that the algorithm is not left to "correct" a potentially insurmountable imbalance in population once the optimal arm has been identified.

We first present a baseline algorithm called *Balanced Exploration* (BE) that nearly achieves the lower bound, but illustrates the key benefit of balancing; this algorithm has the advantage that it needs no knowledge of system parameters. We then use a natural modification of this algorithm called *Balanced Exploration with Arm Elimination* (BE-AE) that achieves the lower bound in Theorem 1, though it uses some knowledge of system parameters in doing so.

## 6.1 Balanced exploration

The BE policy is cautious during the exploration phase in the following sense: it pulls the arm with least accrued reward, to give it further opportunity to ramp up its score just in case its poor performance was bad luck. At the end of the exploration phase, it exploits the empirical best arm for the rest of the horizon.

To define BE, we require an auxiliary sequence $w_k$, $k = 1, 2, \ldots$, used to set the exploration time. The only requirement on this sequence is that $w_k \to \infty$ as $k \to \infty$; e.g., $w_k$ could be $\ln \ln k$ for each postive integer $k$. The BE algorithm is defined as follows.

**Definition 4. Balanced-Exploration (BE) Algorithm:** *Given $T$, let $n = w_T \ln T$.*

1. Exploration phase: *Explore until the (random) time $\tau_n = \min(t : S_b(t) \geq n \ \forall \ b \in A) \wedge T$, i.e., explore until each arm has incurred at least $n$ rewards, while if any arm accrues less than $n$ rewards by time $T$, then $\tau_n = T$. Formally, for $1 \leq t \leq \tau_n$, pull arm $x(t) \in \arg\inf_{a \in A} S_a(t-1)$, with ties broken at random.*

2. Exploitation phase: *Let $\hat{a}^* \in \arg\inf_{a \in A} T_a(\tau_n)$, with tie broken at random. For $\tau_n + 1 \leq t \leq T$, pull the arm $\hat{a}^*$.*

Note that this algorithm only uses prior knowledge of the time horizon $T$, but no other system parameters; in particular, we do not need information on the strength of the positive externality, captured by $\alpha$. Our main result is the following. The proof can be found in the Appendix.

**Theorem 3.** *Suppose $w_k$, $k = 0, 1, 2, \ldots$, is any sequence such that $w_k \to \infty$ as $k \to \infty$. Then the regret of the BE algorithm is as follows:*

*1. If $0 < \alpha < 1$ then $\mathbb{E}[R_T] = O(w_T^\alpha T^{1-\alpha} \ln^\alpha T)$.*

*2. If $\alpha = 1$ then $\mathbb{E}[R_T] = O(w_T \ln^2 T)$.*

*3. If $\alpha > 1$ then $\mathbb{E}[R_T] = O(w_T^\alpha \ln^\alpha T)$.*

In particular, observe that if $w_k = \ln \ln k$, then we conclude $E[R_T] = \tilde{O}(T^{1-\alpha} \ln^\alpha T)$ (if $0 < \alpha < 1$); $E[R_T] = \tilde{O}(\ln^2 T)$ (if $\alpha = 1$); and $E[R_T] = \tilde{O}(\ln^\alpha T)$ (if $\alpha > 1$). Recall that the notation $f(T) = \tilde{O}(g(T))$ implies there exists $k > 0$ such that $f(T) = O(g(T) \ln^k g(T))$.

## 6.2 Balanced exploration with arm elimination

The BE algorithm very nearly achieves the lower bounds in Theorem 1. The additional "inflation" (captured by the additional factor $w_T$) arises in order to ensure the algorithm achieves low regret despite not having information on system parameters.

We now present an algorithm which eliminates the inflation in regret by intelligently eliminating arms that have poor performance during the exploration phase by using upper and lower confidence bounds. The algorithm assumes the knowledge of $T$, $m$, $\alpha$, and $\theta_a$ for each $a$ to the platform (though we discuss the assumption on the knowledge of $\theta_a$ further below). With these informational assumptions, $\lambda_a(t)$ for each $t$ can be computed by the platform. Below, $\hat{\mu}_a(t)$ is an unbiased estimate of $\mu_a$ given observations till time $t$, while $u_a(t)$ and $l_a(t)$ are its upper and lower confidence bounds.

**Definition 5. Balanced Exploration with Arm Elimination (BE-AE) Algorithm:** *Given $T$, $m$, and $\alpha$, as well as $\theta_a$ for each $a \in A$, for each time $t$ and each arm $a$ define:*

$$\hat{\mu}_a(t) = (T_a(t))^{-1} \sum_{k=1}^{t} \frac{X_k}{\lambda_a(k)} \mathbb{1}(I_k = a).$$

*Further, let $c = \min_{a,b \in A} \frac{\theta_a}{m(1+\theta_b)}$. Define $u_a(t) = \hat{\mu}_a(t) + 5\sqrt{\frac{\ln T}{cT_a(t)}}$, and $l_a(t) = \hat{\mu}_a(t) - 5\sqrt{\frac{\ln T}{cT_a(t)}}$.*

*Let $A(t)$ be the set of active arms at time $t$. At time $t = 1$ all arms are active, i.e., $A(1) = A$. At each time $t$ pull arm*

$$I_t \in \arg \inf_{a \in A(t)} S_a(t-1),$$

*with ties broken lexicographically. Eliminate arm $a$ from the active set if there exists an active arm $b \in A(t)$ such that $u_a(t) < l_b(t)$.*

The following theorem shows that the BE-AE algorithm achieves optimal regret, i.e., it meets the lower bounds in Theorem 1. The proof can be found in the Appendix.

**Theorem 4.** *For fixed $m$ and $\alpha$, the regret under the BE-AE algorithm satisfies the following:*

*1. If $0 < \alpha < 1$ then $\mathbb{E}[R_T] = O(T^{1-\alpha} \ln^\alpha T)$.*

*2. If $\alpha = 1$ then $\mathbb{E}[R_T] = O(\ln^2 T)$.*

*3. If $\alpha > 1$ then $\mathbb{E}[R_T] = O(\ln^\alpha T)$.*

As noted above, our algorithm requires some knowledge of system parameters. We briefly describe an approach that we conjecture delivers the same performance as BE-AE, but without knowledge of $\theta_a$ for $a \in A$. Given a small $\epsilon > 0$, first run the exploration phase of the BE algorithm for $n = \epsilon \ln T$ time without removing any arm. For $t$ subsequent to the end of this exploration phase, i.e., once $\epsilon \ln T$ samples are obtained for each arm, we have $N_a(t) = \epsilon \ln T + \theta_a$. Thus, the effect of $\theta_a$ on $\lambda_a(t)$ becomes negligible, and one can approximate $\lambda_a(t)$ by letting $N_b(t) = S_b(t)$ for each arm $b$. We then continue with the BE-AE algorithm as defined above (after completion of the exploration phase). We conjecture the regret performance of this algorithm will match BE-AE as defined above. Proving this result, and more generally removing dependence on $T$, $m$, and $\alpha$, remain interesting open directions.

## 7 Simulations

Below, we summarize our simulation setup and then describe our main findings.

**Simulation setup.** We simulate our model with $m = 2$ arms, with externality strength $\alpha = 1$, arm reward parameters $\mu_1 = 0.5$ and $\mu_2 = 0.3$, and initial biases $\theta_1 = \theta_2 = 1$. For Fig. 1a, we simulate each algorithm one hundred times for each set of parameters. We plot *pseudo-regret* realization from each simulation, i.e., $E[\Gamma_T^*] - \Gamma_T$, where $E[\Gamma_T^*]$ is the expected reward for the Oracle, computed via Monte Carlo simulation, and $\Gamma_T$ is the total reward achieved by the algorithm. Thus, lower pseudo-regret realization implies better performance. For Fig. 1b, each point is obtained by simulating the corresponding algorithm one thousand times. The time horizon $T$ is as mentioned in the figures.

*Parameters for each algorithm.* We simulate UCB($\gamma$) with $\gamma = 3$. For Random-explore-then-commit, we set the exploration time as $\sqrt{T}$ (empirically, this performs significantly better than $\ln T$). For BE, we set $w_T = \beta \ln \ln T$ with $\beta = 2$ (see Definition 4). For BE-AE, cf. Definition 5, we recall that the upper and lower confidence bounds are set as $u_a(t) = \hat{\mu}_a(t) + p\sqrt{\frac{\ln T}{T_a(t)}}$, and $l_a(t) = \hat{\mu}_a(t) - p\sqrt{\frac{\ln T}{T_a(t)}}$ for $p = 5c^{-1/2}$ where $c = \min_{a,b \in A} \frac{\theta_a}{m(1+\theta_b)}$. This choice of $p$ was set in the paper for technical reasons, but unfortunately this choice is suboptimal for finite $T$. The choice of $p = 1/2$ achieves significantly better performance for this experimental setup. The performance is sensitive to small changes in $p$, as the plots illustrate when choosing $p = 5/2$. In contrast, in our experiments, we found that the performance of BE is relatively robust to the choice of $\beta$.

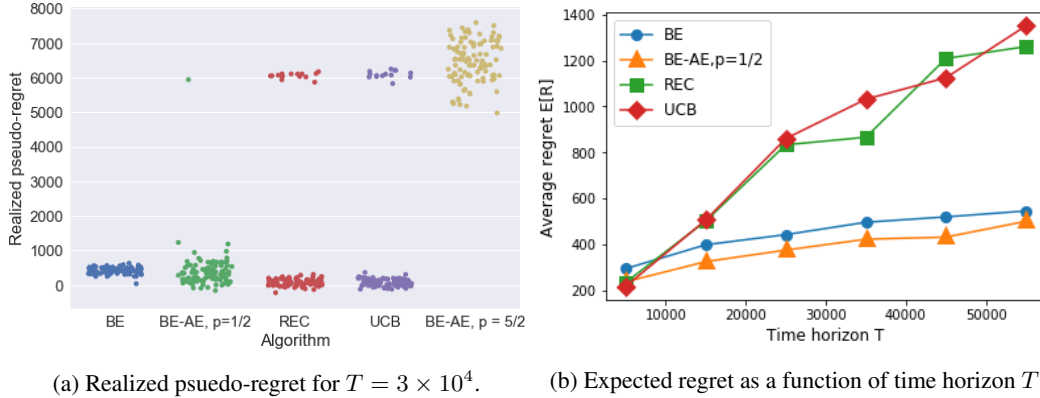

(a) Realized psuedo-regret for $T = 3 \times 10^4$.  (b) Expected regret as a function of time horizon $T$

Figure 1: Performance comparison of algorithms in different parameter regimes. All simulations have $m = 2$ arms, externality strength $\alpha = 1$, arm reward parameters $\mu_1 = 0.5$ and $\mu_2 = 0.3$, and initial arm bias $\theta_1 = \theta_2 = 1$.

**Main findings.** The following are our main findings from the above simulations.

First, even for $\alpha = 1$, REC appears to perform as poorly as UCB. Recall that in Section 5 we show theoretically that the regret is linear for UCB for each $\alpha$, and for REC for $\alpha > 1$. For $\alpha = 1$, we are only able to show that REC exhibits polynomial regret.

Second, for finite $T$, the performance of the (asymptotically optimal) BE-AE algorithm is quite sensitive to the choice of algorithm parameters, and thus may perform poorly in certain regimes. By contrast, the (nearly asymptotically optimal) BE algorithm appears to exhibit more robust performance.

## 8 Discussion and conclusions

It is common that platforms make online decisions under uncertainty, and that these decisions impact future user arrivals. However, most MAB models in the past have decoupled the evolution of arrivals from the learning process. Our model, though stylized by design, provides several non-standard yet interesting insights which we believe are relevant to many platforms. In particular:

1. In the presence of self-reinforcing preferences, there is a cost to being optimistic in the face of uncertainty, as mistakes are amplified.

2. It is possible to mitigate the impact of transients arising from positive externalities by structuring the exploration procedure carefully.

3. Once enough evidence is obtained regarding optimality of a strategy, one may even use the externalities to one's advantage by purposefully shifting the arrivals to a profit-maximizing population.

Of course real-world scenarios are complex and involve other types of externalities which may reverse some of these gains. For example, the presence of negative externalities may preclude the ability to have "all" arrivals prefer the chosen option. Alternatively, arrivals may have "limited memory", so that future arrivals might eventually forget the effect of the externality. Overall, we believe that this is an interesting yet under-explored space of research, and that positive externalities of the kind we study may play a pivotal role in the effectiveness of learning algorithms.

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
