[Supplementary Material]

# 9 Appendix

## 9.1 Proof of Proposition 1

We first show Part 1 of the result. Recall $\lambda_a(t)$ is the probability that the arrival at time $t$ prefers arm $a$, given the past. Thus, from the definition, we have

$$
\begin{aligned}
\mathbb{E}[\Gamma_T^*] &= \mu_{a^*} \mathbb{E}\left[\sum_{t=1}^{T} \lambda_{a^*}(t)\right] \\
&= \mu_{a^*} \mathbb{E}\left[\sum_{t=1}^{T} \frac{N_{a^*}^{\alpha}(t-1)}{N_{a^*}^{\alpha}(t-1)+\theta^{\alpha}}\right] \\
&= \mu_{a^*} \mathbb{E}\left[\sum_{t=1}^{T} \left(1 - \frac{\theta^{\alpha}}{N_{a^*}^{\alpha}(t-1)+\theta^{\alpha}}\right)\right] \\
&= \mu_{a^*} T - \mu_{a^*} \mathbb{E}\left[\sum_{t=1}^{T} \frac{\theta^{\alpha}}{N_{a^*}^{\alpha}(t-1)+\theta^{\alpha}}\right].
\end{aligned}
$$

Using the fact that the maximum reward obtainable at each time is 1, we obtain that $N_a^{\alpha}(t-1) \leq \theta_a + (t-1)$. Thus,

$$
\mathbb{E}[\Gamma_T^*] \leq \mu_{a^*} T - \mu_{a^*} \sum_{t=1}^{T} \frac{\theta^{\alpha}}{(\theta_{a^*}+t-1)^{\alpha}+\theta^{\alpha}},
$$

from which Part 1 follows.

We now show Part 2 of the result. Let $\tau_1 = \inf(t : S_{a^*}(t) = 1)$, i.e., it is the first time instant at which positive reward is obtained. For each $k > 1$ let $\tau_k = \inf(t : S_{a^*}(t) = k) - \tau_{k-1}$, i.e., it represents the time between $(k-1)^{\text{th}}$ and $k^{\text{th}}$ success. By definition, $\tau_k$ has distribution Geometric($\frac{\mu_{a^*} f(k+\theta_{a^*})}{f(k+\theta_{a^*})+\sum_{a\neq a^*} f(\theta_a)}$). One can view $\Gamma_T^*$ as the minimum $n$ such that $\sum_{k=1}^{n+1} \tau_k$ exceeds $T$. Thus, we have

$$
T \leq \mathbb{E}\left[\sum_{k=1}^{\Gamma_T^*+1} \tau_k\right].
$$

Since $\tau_1, \tau_2, \ldots$ is a sequence of independent random variables, and since $\Gamma_T^* + 1$ is a stopping time on this sequence, we obtain the following from Wald's lemma:

$$
\begin{aligned}
T &\leq \mathbb{E}\left[\sum_{k=1}^{\Gamma_T^*+1} \mathbb{E}[\tau_k]\right] \\
&\leq \mathbb{E}\left[\sum_{k=1}^{\Gamma_T^*+1} \frac{f(k+\theta_{a^*})+\sum_{a\neq a^*} f(\theta_a)}{\mu_{a^*} f(k+\theta_{a^*})}\right] \\
&= \mathbb{E}\left[\sum_{k=1}^{\Gamma_T^*+1} \frac{(k+\theta_{a^*})^{\alpha}+\sum_{a\neq a^*} \theta_a^{\alpha}}{\mu_{a^*}(k+\theta_{a^*})^{\alpha}}\right] \\
&= \mathbb{E}\left[\sum_{k=1}^{\Gamma_T^*+1} \left(\frac{1}{\mu_a^*} + \frac{\sum_{a\neq a^*} \theta_a^{\alpha}}{\mu_{a^*}(k+\theta_{a^*})^{\alpha}}\right)\right]
\end{aligned}
$$

Thus we obtain,

$$T \le \frac{1}{\mu_{a^*}}\mathbb{E}[\Gamma_T^* + 1] + \mathbb{E}\left[\sum_{k=1}^{\Gamma_T^*+1} \frac{\sum_{a \neq a^*} \theta_a^\alpha}{\mu_{a^*}(k + \theta_{a^*})^\alpha}\right]$$

$$\le \frac{1}{\mu_{a^*}}\mathbb{E}[\Gamma_T^* + 1] + \mathbb{E}\left[\sum_{k=1}^{T} \frac{\theta^\alpha}{\mu_{a^*}(k + \theta_{a^*})^\alpha}\right]$$

By rearranging we get,

$$\mathbb{E}[\Gamma^* + 1] \ge \mu_{a^*}T - \theta^\alpha \sum_{k=1}^{T} \frac{1}{(k + \theta_{a^*})^\alpha},$$

from which the result follows.

## 9.2 Proof of Theorem 1

We show the result for $\alpha = 1$ and $m = 2$. For other values of $\alpha$ and $m$, the result follows in a similar fashion.

Consider a problem instance where $A = \{a, b\}$, with expected rewards $\mu_a$ and $\mu_b$ respectively. Without loss of generality, assume that $\mu_b < \mu_a < 1$. The rewards obtained by a policy can be simulated as follows. Let $X_{1,a}, X_{2,a}, \ldots$ be a sequence of i.i.d. Bernoulli($\mu_a$) random variables. Similarly, let $X_{1,b}, X_{2,b}, \ldots$ be a sequence of i.i.d. Bernoulli($\mu_b$) random variables. Let $J_t$ represent the set of arms preferred by the arrival at time $t$. Recall that $I_t$ repesents the arm pulled at time $t$. Then the rewards obtained until time $t$, denoted $\Gamma_t$, are given by:

$$\Gamma_t = \sum_{k=1}^{t} \left(\mathbb{1}(I_k = a)\mathbb{1}(a \in J_k)X_{k,a} + \mathbb{1}(I_k = b)\mathbb{1}(b \in J_k)X_{k,b}\right).$$

First, we study the following Oracle, and in particular characterize the maximum payoff achievable. We then use this device to rule out the possibility of policies achieving the performance in the theorem statement.

**Definition 6** (Oracle($t'$)). *Fix time $t'$. The values $\mu_a$, $\mu_b$ are revealed to the Oracle($t'$) after time $t'$.*

**Lemma 1.** *Suppose $t' = o(T)$. Suppose the Oracle($t'$) pulls arm $a$ at all times after $t'$. Then the total expected rewards obtained after time $t'$ by the Oracle is $\mathbb{E}[\Gamma_T - \Gamma_{t'}] = \mu_a(T - t') - O(\mathbb{E}[N_b(t')]\ln T)$.*

*Proof of Lemma 1.* The lemma is analogous to Part (ii) of Proposition 1, with $\theta^\alpha$ replaced by $N_b(t')$, and measuring rewards at times greater than $t'$; thus the lemma can be proved using arguments similar to those used in the theorem. $\square$

The following lemma bounds the payoff achievable by the Oracle after time $t'$.

**Lemma 2.** *Suppose $t' = o(T)$. Any policy used by the Oracle($t'$) satisfies $\mathbb{E}[\Gamma_T - \Gamma_{t'}] = \mu_a(T - t') - \mathbb{E}[N_b(t')]\Omega(\ln T)$.*

*Proof of Lemma 2.* Consider any other policy for the Oracle. Let $\mathcal{U}_a(t')$ be the set of times at which arm $a$ is pulled after $t'$ and the arrival preferred arm $a$: $\mathcal{U}_a(t') = \{t \ge t' : I_t = a, a \in J_t\}$. Let $U_a(t') = |\mathcal{U}_a(t')|$. It is clear that if $U_a(t')$ is $T - t' - \Omega(T)$, then the rewards obtained satisfy $E[\Gamma_T - \Gamma_{t'}] = \mu_a(T - t') - \Omega(T)$. Thus, we assume without loss of generality that after time $t'$, the Oracle follows a policy with $U_a(t') = T - t' - o(T)$.

Using arguments similar to those used in Lemma 6, we obtain:

$$\mathbb{E}[\Gamma_T - \Gamma_{t'}] \le \mu_a U_a(t') - \sum_{t \in \mathcal{U}_a(t')} \frac{\mathbb{E}[N_b(t')]}{t + (t' + \theta_b)} + \mu_b(T - t' - U_a(t')).$$

Since $\mu_b(T - t' - U_a(t')) \le \mu_a(T - t' - U_a(t'))$, we obtain:

$$\mathbb{E}[\Gamma_T - \Gamma_{t'}] \le \mu_a(T - t') - \sum_{t \in \mathcal{U}_a(t')} \frac{\mathbb{E}[N_b(t')]}{t + (t' + \theta_b)}$$

$$\le \mu_a(T - t') - \sum_{t=t'}^{U_a(t')} \frac{\mathbb{E}[N_b(t')]}{t + (t' + \theta_b)}$$

$$= \mu_a(T - t') - \mathbb{E}[N_b(t')]\Omega(\ln U_a(t')).$$

Since $U_a(t') = O(T)$, the lemma follows.

The preceding two lemmas establish that for any $t' = o(T)$, it is asymptotically optimal for the Oracle($t'$) to always pull the best arm after time $t'$. Since the Oracle($t'$) has access to more information, it places a bound on the best achievable regret performance after *any* time $t'$.

Now suppose we are given any policy that has $\mathbb{E}[R_T] = O(\ln^2 T)$. Consider time $t' = T^\gamma$, $\gamma > 0$. For any time $t$ let $\mathcal{T}_a(t) = \{s \le t : I_s = a, a \in J_s\}$; these are the times prior to $t$ when arm $a$ was preferred by the arrival, and was subsequently pulled, and similarly define $\mathcal{T}_b(s) = \{s \le t : I_s = b, b \in J_s\}$. Further, define $\tilde{T}_a(t) = |\mathcal{T}_a(t)|$ and $\tilde{T}_b(t) = |\mathcal{T}_b(t)|$.

Fix a constant $\mu_b'$ such that $\mu_a < \mu_b' < 1$. Consider the following three events, where $c_1 = \frac{1}{2}\frac{\mu_b}{\mu_b'}\gamma$:

$$E_1 := \{N_a(t') \le c_1 \ln T\}; \tag{2}$$
$$E_2 := \{N_a(t') > c_1 \ln T, N_b(t') > c_1 \ln T\}; \tag{3}$$
$$E_3 := \{N_a(t') > c_1 \ln T, N_b(t') \le c_1 \ln T\}. \tag{4}$$

First, note that $R_{t'} = \Omega(1)$ since the Oracle as defined in Section 3.2 is asymptotically optimal. Thus, it suffices to study $\mathbb{E}[R_T - R_{t'}]$.

We trivially have:

$$\mathbb{E}[R_T - R_{t'}] = \mathbb{E}[(R_T - R_{t'})\mathbb{1}(E_1)] + \mathbb{E}[(R_T - R_{t'})\mathbb{1}(E_2)] + \mathbb{E}[(R_T - R_{t'})\mathbb{1}(E_3)].$$

We analyze each of these terms in turn.

Under $E_1$, the total rewards obtained satisfy $\mathbb{E}[\Gamma_T - \Gamma_{t'}] \le \mu_b O(T^\gamma) + \mu_a(T - T^\gamma)$. By our preceding analysis, the Oracle($t'$) obtains reward $\mu_a T - \Theta(\ln T)$ in the same period. Since $\mu_a > \mu_b$, we have that $\mathbb{E}[R_T - R_{t'}|E_1] = \Omega(T^\gamma)$. In particular, this implies that for any policy with $\mathbb{E}[R_T] = O(\ln^2 T)$, we must have $\mathbb{P}(E_1) = o(1)$.

Under $E_2$, we have $E[N_b(t')] \ge c_1 \ln T$. From Lemma 2 we have that $\mathbb{E}[R_T - R_{t'}|E_2] = \Omega(\ln^2 T)$.

Thus, we have that

$$\mathbb{E}[R_T - R_{t'}] \ge \Omega(\ln^2 T)\mathbb{P}(E_2) + \mathbb{E}[R_T - R_{t'}|E_3]\mathbb{P}(E_3),$$

where $\mathbb{P}(E_1) = o(1)$. To conclude the proof, therefore, it suffices to show that $\mathbb{P}(E_3) = o(1)$ as well, since we have that $|\mathbb{E}[R_T - R_{t'}|E_2]| = O(\log^2 T)$ from Lemma 2.

We prove this by considering a modified setting where the reward distribution for arm $a$ is Bernoulli($\mu_a$) (as in the original setting), and where the reward distribution for arm $b$ is Bernoulli($\mu_b'$). Recall, $\mu_a < \mu_b' < 1$. Thus, for the modified setting, arm $b$ is optimal.

We let $\mathbb{P}$ ($\mathbb{E}$) and $\mathbb{P}'$ ($\mathbb{E}'$) denote the probability measure (resp., expectation) corresponding to the original and modified settings, respectively.

It is elementary to show that:

$$\mathbb{P}'(E_3) = \mathbb{E}[\mathbb{1}(E_3)e^{-\hat{K}_{t'}(\mu_b,\mu_b')}]$$

where:

$$\hat{K}_t(\mu_b, \mu_b') = \sum_{s \in \mathcal{T}_b(t)} \left( X_{s,b} \ln \frac{\mu_b}{\mu_b'} + (1 - X_{s,b}) \ln \frac{1 - \mu_b}{1 - \mu_b'} \right).$$

Under the modified setting, again using our analysis of the $\mathsf{Oracle}(t')$, we know the regret incurred conditioned on $E_3$ is $\Omega(T^\gamma)$. Thus for our candidate algorithm we have:

$$O(\ln^2 T) = E[R_T - R_{t'}] \geq \mathbb{P}'(E_3)\Omega(T^\gamma).$$

Thus we obtain $\mathbb{P}'(E_3) = O(T^{-\gamma}\ln^2 T)$. Therefore, $\mathbb{E}[\mathbb{1}(E_3)e^{-\hat{K}_{t'}(\mu_b, \mu_b')}] \leq O(T^{-\gamma}\ln^2 T)$.

But under $E_3$ we have that $\hat{K}_{t'}(\mu_b, \mu_b') \geq c_1 \ln T \ln \frac{\mu_b}{\mu_b'}$, where the right hand side is the value obtained when $X_{t,b}$ for each $t \in \mathcal{T}_b(t')$ is 1. Thus, we get

$$\mathbb{P}(E_3) \leq e^{c_1 \ln T \ln \frac{\mu_b}{\mu_b'}} O(T^{-\gamma}\ln^2 T) = O(T^{c_1 \frac{\mu_b'}{\mu_b} - \gamma}\ln^2 T). \tag{5}$$

But recall that $c_1 = \frac{1}{2}\frac{\mu_b}{\mu_b'}\gamma$. Thus we get $P(E_3) = o(1)$, and in turn, $E[R_T - R_{t'}] = \Omega(\ln^2 T)$, as required.

This completes the proof for $\alpha = 1$. For $0 < \alpha < 1$, following along the lines of Lemma 2, we obtain that any policy used by the $\mathsf{Oracle}(t')$ satisfies $\mathbb{E}[\Gamma_T - \Gamma_{t'}] = \mu_a(T - t') - \mathbb{E}[(N_b(t'))^\alpha]\Omega(T^{1-\alpha})$, and similarly for $0 < \alpha < 1$ we have $E[\Gamma_T - \Gamma_{t'}] = \mu_a(T - t') - \mathbb{E}[N_b(t')^\alpha]\Omega(1)$. Further, for $0 < \alpha < 1$, we set $\gamma > 1 - \alpha$, $c_1 = \frac{1}{2}\frac{\mu_b}{\mu_b'}(\gamma - 1 + \alpha)$ so that bound equivalent to (5) on $\mathbb{P}(E_3)$ for this case is $o(1)$. Rest of the proof follows from arguments similar to that $\alpha = 1$.

### 9.3 Proof of Theorem 2

We first prove the result for the setting with two arms, i.e., $m = 2$, and then generalize later. Suppose $A = \{a, b\}$. Without loss of generality, let $\mu_a > \mu_b$.

Let $\tau_k$ be the time at which arm $a$ is pulled for the $k^{\text{th}}$ time.

Let $Q_k$ be the event that the first $k$ pulls of arm $a$ each saw a user which did not prefer arm $a$.

Let $E_k$ be the event that $\hat{\mu}_b(\tau_k - 1) > \frac{\theta_b \mu_b}{3}$.

Then, under $Q_{k-1} \cap E_{k-1}$, we have the following for each time $t$ s.t. $\tau_{k-1} < t \leq e^{\left(\frac{\theta_b \mu_b}{4}\right)^2 \frac{k-1}{\gamma}}$:

$$u_a(t) < \sqrt{\frac{\gamma \ln e^{\left(\frac{\theta_b \mu_b}{4}\right)^2 \frac{k-1}{\gamma}}}{k-1}} = \frac{\theta_b \mu_b}{4} < \frac{\theta_b \mu_b}{3} < \hat{\mu}_b(t) < u_b(t).$$

Thus, under $Q_{k-1} \cap E_{k-1}$, arm $b$ is pulled for each time $t$ s.t. $\tau_{k-1} < t \leq e^{\left(\frac{\theta_b \mu_b}{4}\right)^2 \frac{k-1}{\gamma}}$, which in turn implies that $\tau_k \geq e^{\left(\frac{\theta_b \mu_b}{4}\right)^2 \frac{k-1}{\gamma}}$.

We now show that there exists an $\epsilon' > 0$ such that $\liminf_{k \to \infty} \mathbb{P}(Q_k \cap E_k) \geq \epsilon'$ from which the result would follow.

Using law of total probability we have,

$$\mathbb{P}(Q_k \cap E_k) \geq \mathbb{P}(Q_{k-1} \cap E_{k-1})\mathbb{P}(Q_k \cap E_k | Q_{k-1}, E_{k-1}).$$

Thus, we have

$$\mathbb{P}(Q_k \cap E_k) \geq \mathbb{P}(Q_{k-1} \cap E_{k-1})\mathbb{P}(E_k | Q_{k-1}, E_{k-1})\mathbb{P}(Q_k | Q_{k-1}, E_{k-1}, E_k). \tag{6}$$

Note that, under $Q_{k-1}$, arm $b$ is pulled at least $k - 1$ times before $\tau_k$. Using standard Chernoff bound techniques it is easy to show that there exists a constant $\delta'$ such that $\mathbb{P}(E_k, E_{k-1} | Q_{k-1}) \geq 1 - e^{-\delta'(k-1)}$. (This can be shown using the standard approach for deriving Chernoff bounds, but with the following version of Markov inequality: $P(X > a, Y > b) \leq E[XY]/(ab)$.) Thus, we get

$$\mathbb{P}(E_k | Q_{k-1}, E_{k-1}) \geq \mathbb{P}(E_k, E_{k-1} | Q_{k-1}) \geq 1 - e^{-\delta'(k-1)}. \tag{7}$$

Under $Q_{k-1} \cap E_{k-1} \cap E_k$, we have that $N_a(\tau_k - 1) = \theta_a$ and

$$N_b(\tau_k - 1) = \theta_b + S_b(\tau_k - 1) = \theta_b + \hat{\mu}_b(\tau_k - 1)T_b(\tau_k - 1).$$

Further, since $\tau_k \geq e^{\left(\frac{\theta_b \mu_b}{4}\right)^2 \frac{k-1}{\gamma}}$, we have

$$T_b(\tau_k - 1) \geq \max\left(k - 1, e^{\left(\frac{\theta_b \mu_b}{4}\right)^2 \frac{k-1}{\gamma}} - k + 1\right).$$

Thus, we have

$$N_b(\tau_k - 1) \geq \theta_b + \frac{\theta_b \mu_b}{3} \max\left(k - 1, e^{\left(\frac{\theta_b \mu_b}{4}\right)^2 \frac{k-1}{\gamma}} - k + 1\right).$$

Thus there exists a constant $c > 0$ such that the following holds for each $k \geq 2$: under $Q_{k-1} \cap E_{k-1} \cap E_k$ we have that

$$N_b(\tau_k - 1) \geq e^{c(k-1)}.$$

Thus, under $Q_{k-1} \cap E_{k-1} \cap E_k$, we have

$$\lambda_a(\tau_k - 1) = \frac{\theta_a}{f(N_b(\tau_k - 1)) + \theta_a} \leq \frac{\theta_a}{f(e^{c(k-1)}) + \theta_a}.$$

Thus, from definition of $Q_k$ we have

$$\mathbb{P}(Q_k | Q_{k-1}, E_{k-1}, E_k) \geq 1 - \frac{\theta_a}{f(e^{c(k-1)}) + \theta_a} = \frac{f(e^{c(k-1)})}{f(\theta_a) + f(e^{c(k-1)})}. \tag{8}$$

Substituting (8) and (7) in (6), we obtain

$$\mathbb{P}(Q_k \cap E_k) \geq \mathbb{P}(Q_{k-1} \cap E_{k-1})\left(1 - e^{-\delta'(k-1)}\right)\left(\frac{f(e^{c(k-1)})}{f(\theta_a) + f(e^{c(k-1)})}\right).$$

Computing recursively, we obtain

$$\mathbb{P}(Q_k \cap E_k) \geq \mathbb{P}(Q_2 \cap E_2)\prod_{l=2}^{k}\left(1 - e^{-\delta'(k-1)}\right)\prod_{l=2}^{k}\left(\frac{f(e^{c(l-1)})}{f(\theta_a) + f(e^{c(l-1)})}\right).$$

Thus, we would be done if we show that $\liminf_{k \to \infty} \sum_{l=2}^{k} \ln\left(1 - e^{-\delta'(k-1)}\right)$ and $\liminf_{k \to \infty} \sum_{l=2}^{k} \ln\left(\frac{f(e^{c(l-1)})}{f(\theta_a) + f(e^{c(l-1)})}\right)$ are both greater than $-\infty$. We show this below. We use the fact that $\ln(1 - x) \geq -x$ for each $x > 0$. We have,

$$\sum_{l=2}^{k} \ln\left(1 - e^{-\delta'(k-1)}\right) \geq -\sum_{l=2}^{k} e^{-\delta'(k-1)},$$

which tends to a constant a $k \to \infty$.

Further,

$$\sum_{l=2}^{k} \ln\left(\frac{f(e^{c(l-1)})}{f(\theta_a) + f(e^{c(l-1)})}\right) \geq -\sum_{l=2}^{k}\left(\frac{f(\theta_a)}{f(\theta_a) + f(e^{c(l-1)})}\right)$$

which tends to a constant a $k \to \infty$ since $f(x)$ is $\Omega\left(\ln^{1+\epsilon}(x)\right)$. This completes the proof for $m = 2$.

For $m > 2$, we can generalize the argument to show that only the worst arm will see non-zero rewards with positive probability by appropriately generalizing the notions of $\tau_k$, $E_k$, and $Q_k$ and arguing along the above lines.

## 9.4 Proof of Proposition 2

We start with a technical result for the algorithm that *indefinitely* pulls arms independently and uniformly at random. For the case where $f$ is linear, we can model the cumulative rewards obtained at each arm via the *generalized Friedman's urn process*. These processes are studied by embedding them into multitype continuous-time Markov branching processes [4, 12], where the expected lifetime of each particle is one at all times.

Here, since we are interested in rewards obtained for more general $f$, we study this by considering multitype branching processes with state-dependent expected lifetimes. For technical reasons, we will assume that $\theta_a$ for each arm $a$ is integer valued and greater than or equal to 1. This allows us to map our problem into an urn type process with initial number of balls of color $a$ in the urn being equal to $\theta_a$. We obtain the following result.

**Proposition 3.** *Suppose that $\theta_a$ for each $a \in A$ is a positive integer. Suppose at each time step $t$ an arm is pulled independently and uniformly at random. The following statements hold:*

(i) *If $f(x) = x^\alpha$ for $0 < \alpha < 1$ then for each $b \neq a^*$, we have that $\frac{N_{a^*}(t)}{N_b(t)} \to \frac{\theta_a}{\theta_b} \left( \frac{\mu_{a^*}}{\mu_b} \right)^{\frac{1}{1-\alpha}}$ almost surely as $t \to \infty$.*

(ii) *If $f(x) = x$ then for each $b \neq a^*$, we have that $\frac{N_{a^*}(t)}{(N_b(t))^{\frac{\mu_{a^*}}{\mu_b}}}$ converges almost surely to a random variable $Y$ with $0 < Y < \infty$ w.p. 1.*

(iii) *If $f(x) = x^\alpha$ for $\alpha > 1$ then there is a positive probability that $N_{a^*}(t)$ is $O(1)$ while for some $b \neq a^*$ we have $N_b(t) \to \infty$ as $t \to \infty$.*

*Proof.* For ease of exposition we will assume that $A = \{a, b\}$. The argument for the more general case is more or less identical.

For now, suppose that $\theta_a = \theta_b = 1$. We will study the process $N = (N_a(t), N_b(t))_{t \in \mathbb{Z}_+}$ by analyzing a multitype continuous time Markov branching process $Z = (Z_a(s), Z_b(s))_{s \in \mathbb{R}_+}$ such that its embedded Markov chain, i.e., the discrete time Markov chain corresponding to the state of the branching process at its jump times, is statistically identical to $N(t)$. By jump time we mean the times at which a particle dies; upon death it may give birth to just one new particle, in which case, the size of the process may not change at the jump times.

We construct $Z$ as follows. Both $Z_a$ and $Z_b$ are themselves independently evolving single dimensional branching processes. Initially, $Z_a$ and $Z_b$ have one particle each, i.e., $|Z_a(0)| = |Z_b(0)| = 1$. Each particle dies at a rate dependent on the size of the corresponding branching processes as follows: at time $s$ each particle of $Z_a$ dies at rate $\frac{f(|Z_a(s)|)}{|Z_a(s)|}$. At the end of its lifetime, the particle belonging to $Z_a$ dies and gives birth to one new particle with probability $\frac{1-\mu_a}{2}$ and two new particles with probability $\frac{\mu_a}{2}$. Similarly for the particles belonging to $Z_b$.

We will use notation $|Z|$ to denote $(|Z_a(s_t)|, |Z_a(s_t)|)$. We now show that the embedded Markov chain of $|Z|$ is statistically identical to $N$. Let $s_1, s_2, \ldots, s_t, \ldots$ represent the jump times of $Z$. We show that the conditional distribution of $N(t)$ given $N(t-1)$ is identical to the conditional distribution of $|Z(s_t)|$ given $|Z(s_{t-1})|$. Since at each time $t$ an arm is chosen at random, we have

$$\mathbb{P}\left( (N_a(t), N_b(t)) = (N_a(t-1)+1, N_b(t)) \big| N(t-1) \right) = \frac{1}{2} \frac{f(N_a(t-1))\mu_a}{f(N_a(t-1)) + f(N_b(t-1))}.$$

Similarly, we can compute the conditional probability for the other values which $N(t)$ can take. Now consider process $Z(\tau)$. After the $(t-1)^{\text{th}}$ jump of $Z$, the rate at which $Z_a$ jumps is $f(|Z_a(s_t)|)$. Thus, the probability that the $(t+1)^{\text{th}}$ jump of $Z$ belongs to $Z_a$ is $\frac{f(|Z_a(s_t)|)}{f(|Z_a(s_t)|) + f(|Z_b(s_t)|)}$. Further, each jump at $Z_a$ results into an increment with probability $\frac{\mu_a}{2}$. Thus we have,

$$\mathbb{P}\left( (|Z_a(s_t)|, |Z_a(s_t)|) = (|Z_a(s_{t-1})|+1|, |Z_a(s_{t-1})|) \big| Z(t-1) \right) = \frac{\mu_a}{2} \frac{f(|Z_a(s_t)|)}{f(|Z_a(s_t)|) + f(|Z_b(s_t)|)}.$$

Further, it is easy to check that $|Z(s_1)|$ and $N(1)$ are identically distributed. Thus, by induction, the embedded Markov chain of $|Z|$ is statistically identical to $N$.

Now, we obtain the following lemma from Theorem 1 in [14]. We say that $f$ is sublinear if there exists $0 < \beta < 1$ such that $f(x) \leq x^{\beta}$.

**Lemma 3.** *If $f(x)$ is linear or sublinear, then*

$$|Z_a(s)| \to w_a(s)(W + o(1)),$$

*where $w_a(s)$ is the inverse function of*

$$g_a(s) = \frac{2}{\mu_a} \int_0^s \frac{1}{f(x)} dx,$$

*and $W$ is a random variable with $0 < W < \infty$ w.p.1. Moreover, $W = 1$ is $f$ is sublinear.*

Now, consider $f(x) = x^{\alpha}$ for $0 < \alpha < 1$. Then, it follows that $w_a(s) = \left( \frac{\mu_a}{s(1-\alpha)} \right)^{\frac{1}{1-\alpha}}$. Thus, we have

$$|Z_a(s)| \left( \frac{2s(1-\alpha)}{\mu_a} \right)^{\frac{1}{1-\alpha}} \to 1 \quad \text{a.s.},$$

and

$$|Z_b(s)| \left( \frac{2s(1-\alpha)}{\mu_b} \right)^{\frac{1}{1-\alpha}} \to 1 \quad \text{a.s..}$$

Thus, part $(i)$ of the theorem follows for the case where $\theta_a = \theta_b = 1$. For general $\theta_a$ and $\theta_b$, we construct as many independent branching processes, apply the above lemma, and the result follows.

Part $(ii)$ follows in a similar fashion and noting that $w_a(s) = e^{\frac{\mu_a}{2}s}$.

We now argue for part $(iii)$. We assume that $\theta_a = \theta_b = 1$, the argument for general $\theta_a$ and $\theta_b$ is similar. We show that if $f(x) = x^{\alpha}$ for $\alpha > 1$ then there exists a time $s < \infty$ such that $\mathbb{P}(|Z_b(s)| = \infty) > 0$. Our result follows from this since for each finite $s$ we have that $\mathbb{P}(Z_a(s) = 1) \geq e^{-s} > 0$. For each $k \geq 1$ let $\gamma_k = \inf\{s \in \mathbb{R}_+ : |Z_b(s)| = k\}$. Clearly, $\gamma_k - \gamma_{k-1}$ is the sum of a random number (with distribution Geometric($\frac{2}{\mu_b}$)) of Exponential($f(k-1)$) distributed random variables. Thus, $\mathbb{E}[\gamma_k] = \frac{2}{\mu_a} \sum_{l=1}^{k-1} \frac{1}{l^{\alpha}}$, which tends to a constant, say $\delta'$, as $k \to \infty$. Thus, $\mathbb{P}(|Z_b(\delta')| = \infty) > 0$. Hence part $(iii)$ follows. This completes the proof of Proposition 3. $\square$

We now continue with proof of Proposition 2. Recall the Definition 3 for Random($\tau$) policy. We assume $\tau = o(T)$, since if not, $E[R_T]$ is $O(T)$ as arms are picked at random during exploration phase.

Part $(iii)$ thus follows from Proposition 3 and noting that $\mathbb{P}(\hat{a}^* \neq a^*)$ is $\Omega(1)$ while the exploitation phase runs for $T - \tau = O(T)$ time.

We now show Part $(ii)$. We first show the following lemma.

**Lemma 4.** *For $\alpha = 1$, under Random($\tau$) policy we have $\mathbb{P}(\hat{a}^* \neq a^*) = \Omega(\tau^{-\frac{\theta_{a^*} \mu_{a^*}}{\mu_b}})$.*

To prove the lemma, for now suppose that $\theta_a = 1$ for each arm $a$. Recall the continuous time Markov-chain branching process construction in the proof of Proposition 3. It is easy to generalize the construction for $m \geq 2$. For general $m$, in process $Z_a(s)$ for each arm $a$ the probability that upon death of a particle it gives birth to two new particles is $\frac{\mu_a}{m}$. For $\alpha = 1$ the process $Z_a(s)$ is a equivalent to the well-known Yule Process [25] and $|Z_a(s)|$ has distribution Geometric($e^{-s\mu_a/m}$) for each $s$. Thus, for each positive real $s$ and positive integer $k$ we have

$$\mathbb{P}(|Z_a(s)| > k) = (1 - e^{-s\mu_a/m})^k.$$

Using $k = \tau$ and $s = \frac{m \ln \tau}{\mu_a}$ we obtain,

$$\mathbb{P}(|Z_a(s)| > \tau) = (1 - e^{-\ln \tau})^{\tau} = (1 - \frac{1}{\tau})^{\tau}$$

Now, let $s' = \sup(s : Z_{a^*}(s) = 0)$. Clearly, $s'$ has Exponential($\frac{\mu_{a^*}}{m}$) distribution. Thus, for arm $b$, we have

$$\mathbb{P}\left(s' > \frac{m \ln \tau}{\mu_b}\right) = e^{-\frac{\mu_{a^*} \ln \tau}{\mu_b}} = \tau^{-\frac{\mu_{a^*}}{\mu_b}}.$$

Now, note that the event $\{s' > \frac{m \ln \tau}{\mu_b}\} \cap \{|Z_b(s)| > \tau\}$ is a subset of the event $S_{a^*}(\tau) = 0$. Thus,

$$\mathbb{P}(\hat{a}^* \neq a^*) \geq \mathbb{P}(s' > \frac{m \ln \tau}{\mu_b}, |Z_b(s)| > \tau) = (1 - \frac{1}{\tau})^\tau \tau^{-\frac{\mu_{a^*}}{\mu_b}} = \Omega(\tau^{-\frac{\mu_{a^*}}{\mu_b}}).$$

Hence, the lemma follows for the case where $\theta_a = 1$ for each arm $a$. For the general values of $\theta_a$, note that we only get an upper bound on $\mathbb{P}(\hat{a}^* \neq a^*)$ if we assume that $\theta_a = 1$ for each $a \neq a^*$. Hence, we assume that $\theta_a = 1$ for each $a \neq a^*$. Then, the lemma follows by the same arguments as above and nothing that $s'$ now has Exponential($\frac{\theta_{a^*}\mu_{a^*}}{m}$) distribution.

We now consider two cases seperately: Case 1 consists of $\tau \leq T^{\frac{\mu_b}{\mu_b + \theta_{a^*}\mu_{a^*}}}$, and Case 2 consists of $\tau \geq T^{\frac{\mu_b}{\mu_b + \theta_{a^*}\mu_{a^*}}}$.

Case 1 ($\tau \leq T^{\frac{\mu_b}{\mu_b + \theta_{a^*}\mu_{a^*}}}$): By Law of Total Expectation, we have

$$\mathbb{E}[R_T] \geq \mathbb{E}[R_T | \hat{a} \neq \hat{a}^*]\mathbb{P}(\hat{a} \neq \hat{a}^*).$$

Since $\tau = o(T)$ we have that $\mathbb{E}[R_T | \hat{a} \neq \hat{a}^*] = O(T)$. Thus,

$$\mathbb{E}[R_T] = \Omega(T)\mathbb{P}(\hat{a} \neq \hat{a}^*) = \Omega(T\tau^{-\frac{\mu_{a^*}}{\mu_b}}),$$

where the last inequality follows from Lemma 4. Since $\tau \leq T^{\frac{\mu_b}{\mu_b + \theta_{a^*}\mu_{a^*}}}$, we have

$$\mathbb{E}[R_T] \geq \Omega(T \times T^{-\frac{\mu_{a^*}}{\mu_b + \theta_{a^*}\mu_{a^*}}}),$$

from which the result follows.

Case 2 ($\tau > T^{\frac{\mu_b}{\mu_b + \theta_{a^*}\mu_{a^*}}}$): Clearly, regret is $\Omega(\tau)$. Thus, we again get $E[R_T] = \Omega(T^{\frac{\mu_b}{\mu_b + \theta_{a^*}\mu_{a^*}}})$, from which the result follows.

This completes the proof of Part $(ii)$.

We now show Part $(i)$. Here again we bound $\mathbb{P}(\hat{a}^* \neq a^*)$ from below by $\mathbb{P}(S_{a^*}(\tau) = 0)$, but we use a more direct approach than considering continuous time branching processes.

**Lemma 5.** *For $0 < \alpha < 1$, there exists a constant $c$ such that under Random($\tau$) policy we have* $\mathbb{P}(\hat{a}^* \neq a^*) \geq e^{-c\left(\tau^{1-\alpha}\right)}$.

Consider an experiment where each arm is pulled at random at each time $t = 1, 2, \ldots, \infty$. Let $\tau_1, \tau_2, \ldots \infty$. be the times at which the reward obtained is 1 while the arm being pulled is either arm $a^*$ or arm $b$. Since arms are pulled at random, we have

$$\mathbb{P}(I_{\tau_1} = b) = \frac{\theta_b^\alpha}{\theta_b^\alpha + \theta_{a^*}^\alpha}.$$

Note that this probability does not depend on the $\theta_a$ for $a \notin \{a^*, b\}$. Similarly, for each $k \geq 1$,

$$\mathbb{P}\left(I_{\tau_{k+1}} = b \middle| \bigcap_{l=1}^k I_{\tau_l} = b\right) = \frac{(\theta_b + k)^\alpha}{(\theta_b + k)^\alpha + \theta_{a^*}^\alpha}.$$

Thus,

$$\mathbb{P}\left(\bigcap_{k=1}^{\tau} I_{\tau_k} = b\right) = \prod_{k=1}^{\tau} \mathbb{P}\left(I_{\tau_k} = b \middle| \bigcap_{l=1}^{k-1} I_{\tau_l} = b\right)$$

$$= \prod_{k=1}^{\tau} \frac{(\theta_b + k - 1)^{\alpha}}{(\theta_b + k - 1)^{\alpha} + \theta_{a^*}^{\alpha}}$$

$$= \prod_{k=1}^{\tau} e^{-\ln \frac{(\theta_b + k - 1)^{\alpha} + \theta_{a^*}^{\alpha}}{(\theta_b + k - 1)^{\alpha}}}$$

$$= e^{-\sum_{k=1}^{\tau} \ln \frac{(\theta_b + k - 1)^{\alpha} + \theta_{a^*}^{\alpha}}{(\theta_b + k - 1)^{\alpha}}}$$

$$= e^{-\sum_{k=1}^{\tau} \ln\left(1 + \frac{\theta_{a^*}^{\alpha}}{(\theta_b + k - 1)^{\alpha}}\right)}$$

$$\geq e^{-\sum_{k=1}^{\tau} \frac{\theta_{a^*}^{\alpha}}{(\theta_b + k - 1)^{\alpha}}}$$

$$\geq e^{-\sum_{k=1}^{\tau} \frac{\theta_{a^*}^{\alpha}}{(1 + k - 1)^{\alpha}}}$$

$$\geq e^{-\Theta\left(\tau^{1-\alpha}\right)}.$$

Under Random($\tau$) policy, the maximum number of successes possible by either arm $a^*$ or $b$ in the exploration phase is $\tau$. Thus, $\mathbb{P}(\bigcap_{k=1}^{\tau} I_{\tau_k} = b)$ as computed above is a lower bound on $P(\hat{a}^* \neq a^*)$. This complete the proof of the lemma.

Similar to $\alpha = 1$, here again we consider two cases: Case 1 consists of $\tau \leq \frac{c}{2\alpha} \ln^{\frac{1}{1-\alpha}} T$, and Case 2 consists of $\tau \geq \frac{c}{2\alpha} \ln^{\frac{\alpha}{1-\alpha}} T$, where $c$ is the constant from Lemma 5.

Case 1 ($\tau < \frac{c}{2\alpha} \ln^{\frac{1}{1-\alpha}} T$): Using argument similar to that for $\alpha = 1$, we have

$$\mathbb{E}[R_T] \geq \Omega(T)\mathbb{P}(\hat{a} \neq \hat{a}^*) = \Omega(Te^{-c\tau^{1-\alpha}}) = \Omega(Te^{-\frac{\alpha}{2} \ln T}) = \Omega(T^{1-\frac{\alpha}{2}}) = \Omega(T^{1-\alpha} \ln^{\frac{\alpha}{1-\alpha}} T),$$

from which the result follows.

Case 2 ($\tau \geq \frac{c}{2\alpha} \ln^{\frac{1}{1-\alpha}} T$): From Part $(i)$ of Proposition 3, as $\tau \to \infty$ we have that $\frac{N_a^{\alpha}(\tau)}{N_{a'}^{\alpha}(\tau)}$ tends to a constant for each pair of arms $a, a'$. Further, $\sum_a N_a(\tau) \leq \tau$. Thus, we have $\mathbb{E}[N_a^{\alpha}(\tau)] = \Omega(\tau^{\alpha})$ for each arm $a$. In other words, there exists a positive constants, say $\beta$, such that $\mathbb{E}[N_a(\tau)] \geq \beta\tau^{\alpha}$ for each $\tau$.

Now consider the exploitation phase. Let $\Gamma'$ be the rewards accrued during this phase. We provide below a bound on $\mathbb{E}[\Gamma']$.

**Lemma 6.** *The rewards accrued during exploitation phase satisfies:*

$$\mathbb{E}[\Gamma'] \leq \mu_{a^*}(T - \tau) - \sum_{t=\tau+1}^{T} \frac{\beta\tau^{\alpha}}{t^{\alpha} + (\tau + \theta_b)^{\alpha}}.$$

The lemma can be shown as follows.

$$\mathbb{E}[\Gamma'] \le \mu_{a^*} \mathbb{E}[\sum_{t=\tau+1}^{T} \lambda_a(t)]$$

$$= \mu_{a^*} \mathbb{E}[\sum_{t=\tau+1}^{T} \frac{N_{a^*}^{\alpha}(t-1)}{\sum_a N_{a^*}^{\alpha}(t-1)}]$$

$$\le \mu_{a^*} \mathbb{E}[\sum_{t=\tau+1}^{T} \frac{N_{a^*}^{\alpha}(t-1)}{N_{a^*}^{\alpha}(t-1) + N_b^{\alpha}(t-1)}]$$

$$= \mu_{a^*} \mathbb{E}[\sum_{t=\tau+1}^{T} \left(1 - \frac{N_b^{\alpha}(t-1)}{N_{a^*}^{\alpha}(t-1) + N_b^{\alpha}(t-1)}\right)]$$

$$= \mu_{a^*}(T-\tau) - \sum_{t=\tau+1}^{T} \mathbb{E}[\frac{N_b^{\alpha}(t-1)}{N_{a^*}^{\alpha}(t-1) + N_b^{\alpha}(t-1)}]$$

$$\le \mu_{a^*}(T-\tau) - \sum_{t=\tau+1}^{T} \mathbb{E}[\frac{N_b^{\alpha}(\tau)}{N_{a^*}(t-1) + N_b^{\alpha}(\tau)}]$$

$$\le \mu_{a^*}(T-\tau) - \sum_{t=\tau+1}^{T} \mathbb{E}[\frac{N_b^{\alpha}(\tau)}{t^{\alpha} + (\tau+\theta_b)^{\alpha}}]$$

$$\le \mu_{a^*}(T-\tau) - \sum_{t=\tau+1}^{T} \frac{\beta\tau^{\alpha}}{t^{\alpha} + (\tau+\theta_b)^{\alpha}}$$

Hence the lemma follows. Further, the maximum rewards accrued during exploration phase if $\mu_{a^*}\tau$. Thus, the overall expected rewards $\mathbb{E}[\Gamma]$ satisfies

$$\mathbb{E}[\Gamma] \le \mu_{a^*}\tau + \mathbb{E}[\Gamma'] \le \mu_{a^*}T - \sum_{t=\tau+1}^{T} \frac{\beta\tau^{\alpha}}{t^{\alpha} + (\tau+\theta_b)^{\alpha}}.$$

Thus, from above inequality and from Proposition 1 we have

$$\mathbb{E}[R_T] = \mathbb{E}[\Gamma^*] - \mathbb{E}[\Gamma] \ge -\Theta(T^{1-\alpha}) + \beta\tau^{\alpha} \sum_{t=\tau+1}^{T} \frac{1}{t^{\alpha} + (\tau+\theta_b)^{\alpha}}$$

Thus, we have

$$\mathbb{E}[R_T] \ge -\Theta(T^{1-\alpha}) + \beta\tau^{\alpha} \sum_{t=\tau+1}^{T} \frac{t^{\alpha} - (\tau+\theta_b)^{\alpha}}{t^{2\alpha} - (\tau+\theta_b)^{2\alpha}}$$

$$= -\Theta(T^{1-\alpha}) + \beta\tau^{\alpha} \sum_{t=\tau+1}^{T} \frac{t^{\alpha} - (\tau+\theta_b)^{\alpha}}{t^{2\alpha}}$$

$$= -\Theta(T^{1-\alpha}) + \beta\tau^{\alpha} \sum_{t=\tau+1}^{T} \frac{1}{t^{\alpha}} - \beta\tau^{\alpha} \sum_{t=\tau+1}^{T} \frac{(\tau+\theta_b)^{\alpha}}{t^{2\alpha}}$$

$$= -\Theta(T^{1-\alpha}) + \beta\tau^{\alpha}\Theta(T^{1-\alpha}) - \Theta(\tau^{2\alpha}T^{1-2\alpha})$$

$$= \Theta(\tau^{\alpha}\Theta(T^{1-\alpha})),$$

where we use $\tau = o(T)$ for the last equality. Recall that we are considering the case where $\tau \ge \frac{c}{2\alpha} \ln^{\frac{1}{1-\alpha}} T$. Note that the above bound takes the smallest value when $\tau = \frac{c}{2\alpha} \ln^{\frac{1}{1-\alpha}} T$. This completes the proof of the theorem.

## 9.5 Proof of Theorem 3

To analyze the BE algorithm we will, as a stepping stone, analyze a slightly more general policy where $n$ is chosen arbitrarily, but still sub-linearly in $T$, as follows.

**Proposition 4.** *Consider a variant of Balanced-Exploration algorithm where $n$ is allowed to be chosen arbitrarily while ensuring that it is $o(T)$. For each $\alpha$, there exists a constant $c_\alpha$ such that the regret under Balanced-Exploration policy satisfies the following:*

1. *If $0 < \alpha < 1$ then regret is $O(n^\alpha T^{1-\alpha} + T e^{-c_\alpha n})$.*

2. *If $\alpha = 1$ then regret is $O(n \ln T + T e^{-c_1 n})$.*

3. *If $\alpha > 1$ then regret is $O(n^\alpha + T e^{-c_\alpha n})$.*

We now prove this proposition, and later use it to prove the theorem.

By Law of Total Expectation, we have

$$
\begin{aligned}
\mathbb{E}[R_T] &= \mathbb{E}[R_T|\hat{a}^* = a^*]\mathbb{P}(\hat{a}^* = a^*) + \mathbb{E}[R_T|\hat{a} \neq \hat{a}^*]\mathbb{P}(\hat{a}^* = a^*) \\
&\leq \mathbb{E}[R_T|\hat{a}^* = a^*] + T\mathbb{P}(\hat{a}^* \neq a^*).
\end{aligned}
\tag{9}
$$

We first obtain a bound on $\mathbb{E}[R_T|\hat{a}^* = a^*]$ and then on $\mathbb{P}(\hat{a}^* \neq a^*)$, from which the proposition would follow.

From the definition of cumulative regret we have

$$
\mathbb{E}[R_T|\hat{a}^* = a^*] = \mathbb{E}[\Gamma_T^*] - \mathbb{E}[\Gamma_T|\hat{a}^* = a^*]
$$

We can lower-bound total rewards obtained by only counting rewards obtained during from time $\tau_n + 1$ to $T$, i.e.,

$$
\mathbb{E}[\Gamma_T|\hat{a}^* = a^*] \geq \mathbb{E}[\Gamma_{\text{exploit}}|\hat{a}^* = a^*],
$$

where $\Gamma_{\text{exploit}}$ represents cumulative rewards obtained during the exploitation phase.

Thus, we get

$$
\mathbb{E}[R_T|\hat{a}^* = a^*] \leq \mathbb{E}[\Gamma_T^*] - \mathbb{E}[\Gamma_{\text{exploit}}|\hat{a}^* = a^*].
\tag{10}
$$

We now obtain a lower bound on $\mathbb{E}[\Gamma_{\text{exploit}}|\hat{a}^* = a^*]$. Note that $N_a(\tau_n) = n + \theta_a$ for each arm $a$. A lower bound on $\mathbb{E}[\Gamma_{\text{exploit}}|\hat{a} = \hat{a}^*, \tau_n]$ is obtained using an argument same as to that used for obtaining the lower bound on $\mathbb{E}[\Gamma^*]$ in Proposition 1, with $\theta^\alpha$ replaced with $\sum_{a \neq a^*}(n + \theta_a)^\alpha$ and looking at times $\tau_n + 1$ to $T$ instead of times $1, \ldots, T$. Thus, we get

$$
\mathbb{E}[\Gamma_{\text{exploit}}|\hat{a}^* = a^*, \tau_n] \geq \mu_{a^*}(T - \tau_n) - \left(\sum_{a \neq a^*}(n + \theta_a)^\alpha\right) \sum_{k=\tau_n}^{T} \frac{1}{(k + \theta_{a^*})^\alpha} - 1.
$$

Taking expectation w.r.t. $\tau_n$, we get

$$
\mathbb{E}[\Gamma_{\text{exploit}}|\hat{a}^* = a^*] \geq \mu_{a^*}(T - \mathbb{E}[\tau_n])
$$

$$
- \left(\sum_{a \neq a^*}(n + \theta_a)^\alpha\right) \mathbb{E}\left[\sum_{k=\tau_n}^{T} \frac{1}{(k + \theta_{a^*})^\alpha}\right] - 1.
$$

$$
\geq \mu_{a^*}(T - \mathbb{E}[\tau_n]) - \left(\sum_{a \neq a^*}(n + \theta_a)^\alpha\right)\left[\sum_{k=1}^{T} \frac{1}{(k + \theta_{a^*})^\alpha}\right] - 1.
$$

Using the above bound and Part 1. of Proposition 1 in (10) we obtain,

$$\mathbb{E}[R|\hat{a}^* = a^*] \leq T\mu_{a^*} - \mu_{a^*}\theta^\alpha \sum_{k=1}^{T} \frac{1}{(\mu_{a^*}k)^\alpha + \theta^\alpha}$$

$$- \mu_{a^*}(T - \mathbb{E}[\tau_n]) + \left(\sum_{a \neq a^*} (n + \theta_a)^\alpha\right) \sum_{k=1}^{T} \frac{1}{(k + \theta_{a^*})^\alpha} + 1.$$

Thus, we obtain

$$\mathbb{E}[R|\hat{a}^* = a^*] \leq \mu_{a^*}\mathbb{E}[\tau_n] - \mu_{a^*}\theta^\alpha \sum_{k=1}^{T} \frac{1}{(\mu_{a^*}k)^\alpha + \theta^\alpha} + \left(\sum_{a \neq a^*} (n + \theta_a)^\alpha\right) \sum_{k=1}^{T} \frac{1}{(k + \theta_{a^*})^\alpha} + 1.$$

We now show that $\mathbb{E}[\tau_n] = O(n)$. During exploration phase, the algorithm operates in $n$ cycles, where at the beginning of cycle $k$ the $N_a$ for each arm $a$ is equal to $k + \theta_a - 1$, and it equals to $k + \theta_a$ at the end of the cycle. Thus, when arm $a$ is pulled, the probability that it obtains a unit reward is at least $\frac{(\theta_a + k - 1)}{\sum_{b \in A}(\theta_b + k)}\mu_a$. Thus, it takes $O(1)$ expected number of attempts on an arm to obtain a unit reward in each cycle. Thus, to obtain $n$ rewards at all arms it takes $\mathbb{E}[\tau_n] = O(n)$ time.

Thus, for $0 < \alpha < 1$ we have

$$\mathbb{E}[R|\hat{a}^* = a^*] \leq \mu_{a^*}O(n) - \Omega(T^{1-\alpha}) + O(n^\alpha T^{1-\alpha})$$
$$= O(n^\alpha T^{1-\alpha}).$$

Similarly we obtain that $\mathbb{E}[R|\hat{a}^* = a^*]$ is $O(n \ln T)$ for $\alpha = 1$ and it is $O(n)$ for $\alpha > 1$.

Thus, the proposition would follow if we show that $\mathbb{P}(\hat{a}^* \neq a^*) \leq e^{-c_\alpha n}$ for some positive constant $c_\alpha$. We show that below. We start with special case where $\theta_a = 1$ for each $a$.

**Lemma 7.** *Suppose $\theta_a = 1$ for each $a \in A$. Let $\delta = \min_{a \neq a^*}(\mu_{a^*} - \mu_a)$. For each arm $b$, there exists a constant $c_b$ independent of $n$ such that*

$$\mathbb{P}\left(\hat{\mu}_b(\tau_n) > \mu_b + \frac{\delta}{2}\right) \leq e^{-c_b n}.$$

*Similarly, there exists a constant $c_b'$ independent of $n$ such that*

$$\mathbb{P}\left(\hat{\mu}_{a^*}(\tau_n) < \mu_{a^*} - \frac{\delta}{2}\right) \leq e^{-c_b'n}.$$

To prove the lemma, note that for each small constant $\epsilon > 0$ there exists an integer constant $k_\epsilon$ such that for each time $t$ after the $k_\epsilon^{\text{th}}$ cycle, we have $(1 - \epsilon)/m \leq \lambda_b(t) \leq (1 + \epsilon)/m$ for each arm $b$. Thus, after a constant $k_{\frac{\delta}{4\mu_b}}$ number of pulls of arm $b$, we have that each pull of arm $b$ results into a success with probability no larger than $\mu_b(1 + \frac{\delta}{4\mu_b})/m$ which equals $\frac{1}{m}(\mu_b + \frac{\delta}{4})$. Thus, when arm $b$ is pulled, time to each success is a Geometric random variable with rate less than or equal to $\frac{1}{m}(\mu_b + \frac{\delta}{4})$. Thus, the first part of the lemma follows from standard exponential concentration result for independent Geometric random variables [10]. Second part of the lemma follows similarly.

Thus, the proposition follows for the case where $\theta_a = 1$ for each $a$. For general values of $\theta_a$ essentially the same argument applies by observing that for each small constant $\epsilon > 0$ there exists an integer constant $k_\epsilon$ such that for each time $t$ after $k_\epsilon^{\text{th}}$ cycle, we have $(1 - \epsilon)/m \leq \lambda_a(t) \leq (1 + \epsilon)/m$ for each arm $a$. Since $k_\epsilon$ does not depend on $n$, the concentration arguments above still hold. This completes the proof for the proposition.

Now, recall that in the statement of Theorem 3 where we set $n = w_T \ln T$. Since $w_k$ is $\omega(k)$, there exists $k$ such that $w_k \geq 2c_\alpha$. Thus, $\mathbb{P}(\hat{a}^* \neq a^*) = O(1/T^2)$. This completes the proof of Theorem 3.

### 9.6 Proof of Theorem 4

We will prove the result for $\alpha < 1$. The result for general $\alpha$ follows using essentially the same argument. Similar to the BE algorithm, the BE-AE algorithm can be thought of as containing

exploration phase and exploitation phase. The exploration phase consists of times $t = 0 \ldots \tilde{t}$ where $\tilde{t} = \max(t \leq T : |A(t)| \geq 2)$, and the exploitation phase consists of times $t > \tilde{t}$. Let the arm active during the exploitation phase be denoted by $\hat{a}^*$. Then, similar to proof of Proposition 4, we have

$$\mathbb{E}[R_T] \leq \mathbb{E}[R_T | \hat{a}^* = a^*] + T\mathbb{P}(\hat{a}^* \neq a^*) \tag{11}$$

$$\leq \sum_{a \neq a^*} \mathbb{E}[T_a(\tilde{t})] + \sum_{a \neq a^*} \mathbb{E}[N_a(\tilde{t})]T^{1-\alpha} + T\mathbb{P}(\hat{a}^* \neq a^*) \tag{12}$$

$$\leq \sum_{a \neq a^*} \mathbb{E}[T_a(T)] + \sum_{a \neq a^*} \mathbb{E}[T_a(T) + \theta_a]T^{1-\alpha} + +T\mathbb{P}(\hat{a}^* \neq a^*) \tag{13}$$

$$\tag{14}$$

Thus, it is sufficient to show that $P(\hat{a}^* \neq a^*) = O(T^{-1})$ and that $\mathbb{E}[T_a(T)] = O(\ln T)$. In turn, it sufficient to show that $\mathbb{P}(\exists t \text{ s.t. } a^* \notin A(t)) = O(T^{-1})$ and that $\mathbb{E}[T_a(T)] = O(\ln T)$. We do that below. We will use the following lemmas, proven in Section 9.6.1.

**Lemma 8.** *We have $\lambda_a(t) \geq c$ for each $t$ and each $a \in A(t)$.*

**Lemma 9.** *For each arm $a \in A$ we have*

1. $\mathbb{P}(\exists t \leq T \text{ s.t. } u_a(t) \leq \mu_a) \leq T^{-1}$

2. $\mathbb{P}(\exists t \leq T \text{ s.t. } l_a(t) \geq \mu_a) \leq T^{-1}$

**Lemma 10.** *Let $\delta = \min_{a \neq a^*}(\mu_{a^*} - \mu_a)$. There exists a constant $\beta$ such that we have*

1. $\mathbb{P}(\exists t \leq T \text{ s.t. } T_a(t) \geq \beta \ln T, u_a(t) \geq \mu_a + \delta/2) \leq T^{-1}$

2. $\mathbb{P}(\exists t \leq T \text{ s.t. } T_a(t) \geq \beta \ln T, l_a(t) \leq \mu_a - \delta/2) \leq T^{-1}$

**Lemma 11.** *Recall $\beta$ from Lemma 10. For a large enough positive constant $\gamma$ we have that for $t' = \gamma \ln T$ we have $\mathbb{P}(T_a(t') \leq \beta \ln T, a \in A(t')) \leq T^{-2}$ for each arm $a \in A$.*

Now, using union bound we get,

$$\mathbb{P}(\exists t \text{ s.t. } a^* \notin A(t)) \leq \sum_{a \neq a^*} \mathbb{P}(\exists t \text{ s.t. } u_{a^*}(t) < l_a(t))$$

$$\leq \sum_{a \neq a^*} (\exists t \text{ s.t. } \mathbb{P}(u_{a^*}(t) \leq \mu_{a^*}) + \mathbb{P}(\exists t \text{ s.t. } l_a(t) \geq \mu_a))$$

$$= O(1/T),$$

where the last bound follows from Lemma 9. Thus, it is now sufficient to show that $\mathbb{E}[T_a(T)]$ for each $a \neq a^*$ is $O(\ln T)$. Let $\gamma > 0$ be a constant to be determined. Let $t' = \gamma \ln T$. We have,

$$\mathbb{E}[T_a(T)] \leq \mathbb{E}[T_a(T) | a \notin A(t')] + T\mathbb{P}(a \in A(t'))$$

$$\leq t' + T\mathbb{P}(a \in A(t'))$$

$$= \gamma \ln T + T\mathbb{P}(a \in A(t'))$$

Thus, we will be done if we show that $\mathbb{P}(a \in A(t')) = O(T^{-1})$ for a large enough $\gamma$. We do that below. By Law of Total Probability and the fact that $\mathbb{P}(\exists t \text{ s.t. } a^* \notin A(t)) = O(T^{-1})$ as shown above, we have

$$\mathbb{P}(a \in A(t')) \leq \mathbb{P}(a \in A(t'), a^* \in A(t')) + \mathbb{P}(a^* \notin A(t')) = \mathbb{P}(a \in A(t'), a^* \in A(t')) + O(T^{-1}).$$

Further,

$$\mathbb{P}(a \in A(t'), a^* \in A(t')) \leq \mathbb{P}(T_a(t') \leq \beta \ln T, a \in A(t')) + \mathbb{P}(T_{a^*}(t') \leq \beta \ln T, a^* \in A(t'))$$

$$+ \mathbb{P}(a \in A(t'), a^* \in A(t'), T_a(t') \geq \beta \ln T, T_{a^*}(t') \geq \beta \ln T)$$

$$= O(1/T^2) + \mathbb{P}(a \in A(t'), a^* \in A(t'), T_a(t') \geq \beta \ln T, T_{a^*}(t') \geq \beta \ln T),$$

where the last equality follows from Lemma 11. We now provide a bound on the last term of the above inequality. Event $a, a^* \in A(t')$ implies that $u_a(t') < l_{a^*}(t')$. Thus, we get

$$\mathbb{P}(a \in A(t'), a^* \in A(t'), T_a(t') \geq \beta \ln T, T_{a^*}(t') \geq \beta \ln T)$$
$$\leq \mathbb{P}(u_a(t') < l_{a^*}(t'), T_a(t') \geq \beta \ln T, T_{a^*}(t') \geq \beta \ln T),$$
$$\leq \mathbb{P}(u_a(t') < \mu_a + \delta/2, T_a(t') \geq \beta \ln T) + \mathbb{P}(l_{a^*}(t'), T_{a^*}(t') \geq \beta \ln T),$$
$$= O(1/T),$$

where the last inequality follows from Lemma 10. Hence the result follows.

### 9.6.1 Proof of lemmas used in proof of Theorem 4

**Lemma 8.** We have $\lambda_a(t) \geq c$ for each $t$ and each $a \in A(t)$.

*Proof.* From the definition of the algorithm, we have $|S_a(t) - S_b(t)| \leq 1$ for each $a, b \in A(t)$. Further, for each $a \in A(t)$ and $b \notin A(t)$ we have $S_a(t) \geq S_b(t) - 1$. Let $b' \in \arg\max_a \theta_a$. Thus, for each $a \in A(t)$ we have

$$\lambda_a(t) = \frac{S_a(t) + \theta_a}{\sum_b (S_b(t) + \theta_b)} \geq \frac{S_a(t) + \theta_a}{\sum_b (S_a(t) + 1 + \theta_b)} \geq \frac{S_a(t) + \theta_a}{m(S_a(t) + 1 + \theta_{b'})} \geq \frac{\theta_a}{m(1 + \theta_{b'})}.$$

Hence, the lemma holds since $c = \min_a \frac{\theta_a}{m(1+\theta_{b'})}$. $\square$

**Lemma 9.** For each arm $a \in A$ we have

1. $\mathbb{P}(\exists t \leq T \text{ s.t. } u_a(t) \leq \mu_a) \leq T^{-1}$
2. $\mathbb{P}(\exists t \leq T \text{ s.t. } l_a(t) \geq \mu_a) \leq T^{-1}$

*Proof.* We first prove the first part 1. We will use the following result, known as Freedman's inequality for martingales.

**Theorem 5** (Freedman [11]). *Let $(W_t, \mathcal{F}_t)_{i=0,..,T}$ be a real valued martingale. Let $(\xi_t, \mathcal{F}_t)_{t=0,..,T}$ be the sequence of corresponding martingale differences, i.e., $W_t = \sum_{i=0}^{t} \xi_t$, s.t. $\xi_0 = 0$. Let $V_k = \sum_{i=1}^{t} \mathbb{E}[\xi^2 | \mathcal{F}_{i-1}]$. Suppose $\xi_t \leq \epsilon$ for a some positive $\epsilon$. Then, the following holds for all positive $w$ and $v$.*

$$\mathbb{P}\left(\exists t \text{ s.t. } W_t \geq w \text{ and } V_t \leq v\right) \leq \exp\left(-\frac{w^2}{2(v + w\epsilon)}\right).$$

Let $M_0 = 0$ and for each $t \geq 1$ let $M_t = \mu_a T_a(t) - \hat{\mu}_a(t) T_a(t)$. Let $\{\mathcal{F}_t\}_{t \geq 0}$ represent the filtration where $\mathcal{F}_0 = \{\emptyset, \Omega\}$ and $\mathcal{F}_t$ captures what is known to the platform at each time $t$. Then, it is easy to check that $(M_t, \mathcal{F}_t)_{i=0,...,T}$ forms a martingale. Consider stopping times $\tau_k$ where $\tau_k = \inf\{t : T_a(t) = k\}$ if $T_a(T) \leq k$ else $\tau_k = T$. Let $Y_0 = 0$ and $Y_k = M_{\tau_k}$. From Optional Sampling Theorem (see Chapter A14 in [24]) we get that $(Y_k, \mathcal{F}_{\tau_k})_{k \geq 0}$ is a martingale.

We now provide bound on $\mathbb{P}(\exists t \text{ s.t. } u_a(t) \leq \mu_a)$. Note that if $T_a(t) < 25c^{-1} \ln T$ then $5\sqrt{\frac{\ln T}{cT_a(t)}} > 1 \geq \mu_a$ and $u_a(t) > \mu_a$. Thus, we get,

$$\mathbb{P}(\exists t \text{ s.t. } u_a(t) \leq \mu_a) = \mathbb{P}\left(\exists t \text{ s.t. } T_a(t) \geq 25c^{-1} \ln T, \hat{\mu}_a(t) + 5\sqrt{\frac{\ln T}{cT_a(t)}} \leq \mu_a\right)$$

$$= \mathbb{P}\left(\exists t \text{ s.t. } T_a(t) \geq 25c^{-1} \ln T, M_t \geq 5T_a\sqrt{\frac{\ln T}{cT_a(t)}}\right)$$

$$= \mathbb{P}\left(\exists k \geq 25c^{-1} \ln T \text{ s.t. } Y_k \geq 5\sqrt{c^{-1}k \ln T}\right)$$

Let $D_k = Y_k - Y_{k-1} = \mu_a - \frac{X_{\tau_k}}{\lambda_a(\tau_k)}$. From Lemma 8 we have $-c^{-1} \leq D_k \leq \mu_a \leq 1$ for each $k$.

From the definition of BE-AE algorithm, since ties are broken deterministically, we have that $I_t \in \mathcal{F}_{t-1}$. Thus, $\tau_k - 1$ is a stopping time. Thus, $\mathcal{F}_{\tau_k-1}$ is well defined. Now, $\mathbb{E}[D_k^2|\mathcal{F}_{\tau_k-1}] = \mathbb{E}[\mathbb{E}[D_k^2|\mathcal{F}_{\tau_k-1}]|\mathcal{F}_{\tau_k-1}]$, where

$$\mathbb{E}[D_k^2|\mathcal{F}_{\tau_k-1}] = \frac{\mu_a}{\lambda_a(\tau_k)} - \mu_a^2 \le \frac{1}{\lambda_a(\tau_k)} \le c^{-1}.$$

Thus for each $k$ we have $\sum_{i=1}^k \mathbb{E}[D_i^2|\mathcal{F}_{\tau_{i-1}}] \le c^{-1}k$ with probability $t$.

Fix a $k$ such that $25c^{-1}\ln T \le k \le T$. Using Freedman's inequality we get

$$\mathbb{P}\left(\exists i \text{ s.t. } Y_i \ge 5\sqrt{c^{-1}k\ln T}, \sum_{j=1}^i \mathbb{E}[D_j^2|\mathcal{F}_{\tau_{j-1}}] \le c^{-1}k\right) = \exp\left(-\frac{25kc^{-1}\ln T}{2(c^{-1}k + 5\sqrt{c^{-1}k\ln T})}\right)$$
$$\le \exp\left(-2c^{-1}\ln T)\right)$$
$$\le T^{-2}$$

Thus,

$$\mathbb{P}\left(Y_k \ge 5\sqrt{c^{-1}k\ln T}, \sum_{j=1}^k \mathbb{E}[D_j^2|\mathcal{F}_{\tau_{j-1}}] \le c^{-1}k\right) \le T^{-2}.$$

But, as saw above, $\sum_{j=1}^k \mathbb{E}[D_j^2|\mathcal{F}_{\tau_{j-1}}] \le c^{-1}k$ holds w.p. 1. Thus,

$$\mathbb{P}\left(Y_k \ge 5\sqrt{c^{-1}k\ln T}\right) \le T^{-2}.$$

Using union bound, we get

$$\mathbb{P}\left(\exists k \ge 25c^{-1}\ln T \text{ s.t. } Y_k \ge 5\sqrt{c^{-1}k\ln T}\right) \le T^{-1}.$$

Hence, we get $\mathbb{P}(\exists t \text{ s.t. } u_a(t) \le \mu_a) \le T-1$. Proof for part 2. is similar to above, except that we work with martingale $(-Y_k, \mathcal{F}_{\tau_k})_{k\ge 0}$ instead of $(Y_k, \mathcal{F}_{\tau_k})_{k\ge 0}$.  □

**Lemma 10.** Let $\delta = \min_{a \ne a^*}(\mu_{a^*} - \mu_a)$. There exists a constant $\beta$ such that we have

1. $\mathbb{P}(\exists t \le T \text{ s.t. } T_a(t) \ge \beta \ln T, u_a(t) \ge \mu_a + \delta/2) \le T^{-1}$
2. $\mathbb{P}(\exists t \le T \text{ s.t. } T_a(t) \ge \beta \ln T, l_a(t) \le \mu_a - \delta/2) \le T^{-1}$

*Proof.* We first prove the second part. Arguing along the lines similar to the proof of Lemma 9, it is sufficient to show that there exists $\beta$ such that

$$\mathbb{P}(\exists k \ge \beta \ln T \text{ s.t. } Y_k \ge \delta k/2 - 5\sqrt{c^{-1}k\ln T}) \le T^{-2}.$$

For large enough $\beta$, for each $k \ge \beta \ln T$ we have $\delta k/2 - 5\sqrt{c^{-1}k\ln T} \ge \delta k/4$. Further, for large enough $k$, using Freedman's inequality and the arguments similar to those in Lemma 9, we get

$$\mathbb{P}(Y_k \ge \delta k/4) \le \exp(-\frac{\delta^2 k^2/16}{2(c^{-1}k + c^{-1}\delta k/4)}) = \exp(-c_1 k)$$

where $c_1 > 0$. For large enough $\beta$ we have $c_1 k \ge 2\ln T$ for each $k \ge \beta \ln T$, and thus $\mathbb{P}(Y_k \ge \delta k/4) \le T^{-2}$. The result then follows by using a union bound. The second part of the result follows in a similar fashion by using martingale $(-Y_k, \mathcal{F}_{\tau_k})_{k\ge 0}$ instead of $(Y_k, \mathcal{F}_{\tau_k})_{k\ge 0}$.  □

**Lemma 11.** Recall $\beta$ from Lemma 10. For a large enough positive constant $\gamma$ we have that for $t' = \gamma \ln T$ we have $\mathbb{P}(T_a(t') \le \beta \ln T, a \in A(t')) \le T^{-2}$ for each arm $a \in A$.

*Proof.* Let $\Delta = \min_a \mu_a$. Thus, $\mathbb{P}(X_t = 1) \geq c\Delta$ for each $t$ under the BE-AE algorithm. Thus, using standard Chernoff bound, for a large enough $\gamma$ we have $\mathbb{P}(\sum_{t=1}^{t'} X_t \leq m\beta \ln T + m) \leq e^{-2\ln T}$. Since under BE-AE we have $|S_b(t) - S_{b'}(t)| \leq 1$ for each $b, b' \in A(t)$, for a large enough $\gamma$, $\mathbb{P}(\exists a \in A(t') \text{ s.t. } S_a(t') \leq \beta \ln T) \leq e^{-2\ln T}$. Thus, $\mathbb{P}(\exists b \in A(t') \text{ s.t. } T_b(t') \leq \beta \ln T) \leq e^{-2\ln T}$. Hence the lemma holds. $\qquad\square$