[Reviews · NeurIPS 2018]

Reviewer 1



The paper studies the interesting problem of learning with externalities, in a multi-armed bandit (MAB) setting. The main idea is that there might be a bias in the preferences in the users arriving on on-line platforms. Specifically, future user arrivals on the on-line platforms are likely to have similar preferences to users who have previously accessed the same platform and were satisfied with the service. Since some on-line platforms use MAB algorithms for optimizing their service, the authors propose the Balanced Exploration (BE) MAB algorithm, which has a structured exploration strategy that takes into account this potential "future user preference bias" (referred to as "positive externalities"). The bias in the preference of the users is translated directly into reward values specific to users arriving to on-line platform: out of the $m$ possible items/arms, each user has a preference for a subset of them (the reward for this being a Bernoulli reward with mean proportional to the popularity of the arm) and the rewards of all other arms will always be null. An analysis of the asymptotic behavior of the BE algorithm is given, as well as a comparison to the behavior of UCB and explore-then-exploit MAB strategies. The paper is well written and the studied problem is interesting. The proofs seem sound. I also appreciate the analysis of the cost of being optimistic in the face of uncertainty (as in standard MAB strategies) when they hypothesis of having positive externalities is verified. On the other hand, the motivation of the paper comes from real-world applications and the paper contains no empirical evaluation. It would have liked to observe the behavior of the algorithm on a dataset coming from platforms cited in the introduction of the paper or on some synthetic data. This would have been helpful to have a more thorough view on the practical significance of the contribution and on the type of bias that appears in user's preferences and its impact on the performance measure of recommendations/allocation strategies. ======================= Post author response edit: I thank the authors for their response and clarifications, I also appreciate the added simulation results.

Reviewer 2



Summary of the paper: The paper deals with a setting of stochastic multi-armed bandits with positive externalities. The setting is reasonably novel and the same is well motivated with the practical applications. First, they have derived lower bounds on the expected regret of any policy in the setting considered. Second, they have shown that the classic UCB algorithm need not perform well in the considered setting and argued that it can incur linear regret. Then, they have considered another policy called Random-Explore-Then-Commit and showed that it is better than UCB in some cases but it can still incur linear regrets. Next, they proposed a policy called Balance-Exploration which does not require any system parameters and showed that this policy's performance is near to the achievable lower bounds. Finally, they proposed a policy called Balance-Exploration-with-Arm-Elimination which requires few system parameters and proved that this policy is indeed optimal. Overall, the paper is well written and easy to follow. Providing some simulations would have been better in order to illustrate the linear regret behaviour of UCB and Random-Explore-Then-Commit policies. It would be great if they could provide proof sketches for their key results at least. There are few typos in the appendix. For instance, the second equation in the Section 8.1 should be an inequality, and $\alpha$ should not be there in the following inequality, $N^{\alpha}_a(t-1) \leq \theta_a + (t - 1),$ in the same section.

Reviewer 3



In this paper, the authors study the effects of positive externalities in MAB. In real platforms such as news site, the positive externalities come from the fact that if one of the arms generates rewards, then the users who prefer that arm become more likely to arrive in the future. If the platform knows the user preferences or could infer it from data, the usual way for handling positive externalities is to use contextual bandits: the services are personalized for each kind of users. However, most of the time the contexts are not very informative about the user preferences. The usual alternative is to consider that the reward process evolves during time, and hence to use adversarial bandits or piecewise stationary bandits. Here, the authors propose to take advantage of prior knowledge on how the reward process evolves during time: the positive externalities. The positive externalities change the usual trade-off between exploration and exploitation. Indeed, the effects of the choices of the platform are amplified. If the platform chooses the optimal arm, then this choice is amplified by the arriving of the users that like the optimal arm. However, if the algorithm chooses a sub-optimal arm, the price to pay in terms of future rewards can be dramatic. In order to analyze the positive externality effect, the authors introduce the regret against an Oracle which knows the optimal arm. Depending on the value of \alpha, which measures the strength of positive externalities, a regret lower bound of MAB with positive externalities is provided. Then they bring out that classical approaches are suboptimal. Firstly, they show that UCB algorithm achieves a linear regret for bandits with positive externalities. Secondly, they show that explore then exploit algorithm may incur a linear regret when \alpha > 1. A first algorithm called Balanced Exploration (BE) is introduced. In the exploration phase the arm which has the lowest cumulated reward is chosen, while in the exploitation phase the arm which has been the less chosen in the exploration phase is played. it is worth noting that BE algorithm needs only to know the time horizon. The analysis of this algorithm shows that it is near optimal. A second algorithm, that assumes that in addition to the time horizon the parameters of positive externalities are known (\alpha and \theta_a), is proposed. This algorithm uses an unbiased estimator of the mean reward thanks to the knowledge of the arrival probability. The algorithm plays the arm that has received the least rewards in order to explore. In order to exploit, the suboptimal arms are eliminated when their upper bounds are lesser that the lower bound of the estimated best arm. The authors show that Balanced Exploration with Arm Elimination is optimal. I regret that there are no experiments because it could be interesting for the reader to observe the gap in performances between BE, where the arrival probabilities are unknown and BE-AE, where the arrival probabilities are known. Overall, the paper is very interesting and I vote for acceptance. Minor Comments: The notation \theta^\alpha line 151 is misleading. You use the notation power alpha for N_a^\alpha and for \theta^\alpha but it does not mean the same thing. I suggest to use \theta(\alpha). Line 196, there is a typo u_a(0) = \infty and not 0.